# Statistical Optimal Transport posed as Learning Kernel Mean Embedding

**J. Saketha Nath**
Department of Computer Science and Engineering,
Indian Institute of Technology Hyderabad, INDIA.
saketha@cse.iith.ac.in

**Pratik Jawanpuria**
Microsoft IDC,
Hyderabad, INDIA.
pratik.jawanpuria@microsoft.com

## Abstract

The objective in statistical Optimal Transport (OT) is to consistently estimate the optimal transport plan/map solely using samples from the given source and target marginal distributions. This work takes the novel approach of posing statistical OT as that of learning the transport plan's kernel mean embedding from sample based estimates of marginal embeddings. The proposed estimator controls over-fitting by employing maximum mean discrepancy based regularization, which is complementary to $\phi$-divergence (entropy) based regularization popularly employed in existing estimators. A key result is that, under very mild conditions, $\epsilon$-optimal recovery of the transport plan as well as the Barycentric-projection based transport map is possible with a sample complexity that is completely dimension-free. More-over, the implicit smoothing in the kernel mean embeddings enables out-of-sample estimation. An appropriate representer theorem is proved leading to a kernelized convex formulation for the estimator, which can then be potentially used to perform OT even in non-standard domains. Empirical results illustrate the efficacy of the proposed approach.

## 1   Introduction

Optimal Transport is proving to be an increasingly successful tool in solving diverse machine learning problems. Recent research shows that variants of Optimal Transport (OT) achieve state-of-the-art performance in various machine learning (ML) applications such as data alignment/integration [2, 24, 42, 19], domain adaptation [7, 34], model interpolation/combination [38, 35, 9], natural language processing [43, 44] etc. It is also shown that OT based (Wasserstein) metrics serve as good loss functions in both supervised [13, 20] and unsupervised [16] learning.

Given two marginal distributions over source and target domains, and a cost function between elements of the domains, the classical OT problem (Kantorovich's formulation) is that of finding the joint distribution whose marginals are equal to the given marginals, and which minimizes the expected cost with respect to this joint distribution [22]. This joint distribution is known as the (optimal) transport plan or the optimal coupling. A related object of interest for ML applications is the so-called Barycentric-projection based transport map corresponding to a transport plan (e.g., refer Equation (11) in [37]). Though OT techniques already improve state-of-the-art in many ML applications, there are two main bottlenecks that seem to limit OT's success in ML settings:

- while continuous distributions are ubiquitous, algorithms for finding the transport plan/map over continuous domains are very scarce [15]. The situation is worse in case of non-standard domains, which are not uncommon in ML.
- the marginal distributions are never available, and merely samples from them are given. The variant of OT where the transport plan/map needs to be estimated merely using samples from the marginals is known as the statistical OT problem. Unfortunately, this estimation problem is plagued with the

curse of dimensionality: the sample complexity of $O(m^{-1/d})$, where $m$ is number of samples, and $d$ is the dimensionality of data, cannot be improved without further assumptions [29].

Though several works alleviated the curse of dimensionality [14, 29, 18], none of them completely remove the adversarial dependence on dimensionality. Further, authors in [15, 14, 12, 4] comment that estimators that are free from the curse of dimensionality are important, yet not well-studied. The concluding report from a recent workshop on OT (refer section 2 in [4]) summarizes that one of the major open problems in this area is to design estimators in context of continuous statistical OT whose sample complexity is not a strong function of the dimension (ideally dimension-free).

Our work focuses on this challenging and important problem of statistical OT over continuous domains, and seeks consistent estimators for $\epsilon$-optimal transport plan/map, whose sample complexity is dimension-free. To this end, we take the novel approach of equivalently re-formulating the statistical OT problem solely in terms of the relevant kernel mean embeddings [26]. More specifically, our formulation finds the (characterizing) kernel mean embedding of a joint distribution with least expected cost, and whose marginal embeddings are close to the given-sample based estimates of the marginal embeddings. There are several advantages of this new approach:

1. because the samples based estimates of the kernel mean embeddings of the marginals are known to have sample complexities that are dimension-free, it is expected that the sample complexity remains dimension-free even for the proposed estimator of the transport plan embedding.
2. kernel embeddings provide implicit smoothness, as controlled by the kernel. Appropriate smoothness not only improves the quality of estimation, but also enable out-of-sample estimation.
3. since Maximum Mean Discrepancy (MMD) is the natural notion of distance in the kernel mean embedding space, this reformulation facilitates MMD based regularization for controlling overfitting. Such regularizers are complementary to the $\phi$-divergence (or entropy) based regularizers popularly employed in existing estimators. [40] observe that MMD and $\phi$-divergence based regularization exhibit complementary properties and hence both are interesting to study.

A key result from this work is that, under very mild conditions, the proposed methodology can recover an $\epsilon$-optimal transport plan and corresponding (Barycentric-projection based) transport map with a sample complexity, $O(m^{-1/2})$, which is completely[1] dimension-free. Another contribution is an appropriate representer theorem that guarantees finite characterization for the transport plan embedding, leading to a fully kernelized and convex formulation for the estimation. Thus the same formulation can potentially be used for obtaining estimators with all variants of OT: continuous, semi-discrete, and discrete, merely by switching the kernel between the Kronecker delta and the Gaussian kernels. More importantly, the same can be used to solve OT problems in non-standard domains using appropriate universal kernels [5]. Finally, we discuss special cases where the proposed convex formulation can be solved efficiently using ADMM based solver [3]. Empirical results on synthetic and real-world datasets illustrate the efficacy of the proposed approach. The proofs of all the theorems discussed in this paper are provided in the technical report [28].

## 2   Background on Optimal transport and Kernel Embeddings

In this section we briefly summarize the theories of optimal transport (OT) and kernel mean embeddings, which are essential for understanding the proposed methodology.

**Optimal Transport**

We begin with a brief discussion on OT. For more details, please refer [33], which is a comprehensive monologue on the subject with focus on recent developments related to machine learning.

Let $\mathcal{X}, \mathcal{Y}$ be any two sets that form locally compact Hausdorff topological spaces. We denote the set of all Radon probability measures over $\mathcal{X}$ by $\mathcal{M}^1(\mathcal{X})$; whereas we denote the set of strictly positive measures by $\mathcal{M}^1_+(\mathcal{X})$. Let $c : \mathcal{X} \times \mathcal{Y}$ denote a function that evaluates the cost between elements in $\mathcal{X}, \mathcal{Y}$ and let $p_s \in \mathcal{M}^1_+(\mathcal{X}), p_t \in \mathcal{M}^1_+(\mathcal{Y})$. Then, the Kantorovich's OT formulation [22] is:

$$
\begin{aligned}
\min_{\pi \in \mathcal{M}^1(\mathcal{X}, \mathcal{Y})} \quad & \int c(x, y) \, \mathrm{d}\pi(x, y), \\
\text{s.t.} \quad & \pi^{\mathcal{X}} = p_s, \pi^{\mathcal{Y}} = p_t,
\end{aligned}
\tag{1}
$$

where $\pi^{\mathcal{X}}, \pi^{\mathcal{Y}}$ denote the marginal measures of $\pi$ over $\mathcal{X}, \mathcal{Y}$ respectively. An optimal solution of (1) is referred to as an optimal transport plan or optimal coupling.

**Statistical OT**: In the setting of statistical OT, the marginals $p_s, p_t$ are not available; however, iid samples from them are given. Let $\mathcal{D}_x = \{x_1, \dots, x_m\}$ denote the set of $m$ iid samples from $p_s$ and let $\mathcal{D}_y = \{y_1, \dots, y_n\}$ denote $n$ iid samples from $p_t$. The cost function is known only at the sample data points. Let $\mathcal{C} \in \mathbb{R}^{m \times n}$ denote the cost matrix with with $(i,j)^{th}$ entry as $c(x_i, y_j)$.

A popular way to estimate the optimal plan in (1) is to employ the sample based estimates for the marginals: $\hat{p}_s \equiv \frac{1}{m} \sum_{i=1}^{m} \delta_{x_i}$ and $\hat{p}_t \equiv \frac{1}{n} \sum_{j=1}^{n} \delta_{y_j}$, in place of the true (unknown) marginals. Here, $\delta$ denotes the Dirac delta function. In such a case, (1) simplifies as the standard discrete OT problem:

$$\begin{aligned} \min_{\pi \in \mathbb{R}^{m \times n}} \quad & tr(\pi \mathcal{C}^{\top}), \\ \text{s.t.} \quad & \pi \mathbf{1} = \tfrac{1}{m}\mathbf{1}, \pi^{\top}\mathbf{1} = \tfrac{1}{n}\mathbf{1}, \pi \geq \mathbf{0}, \end{aligned} \quad (2)$$

where $tr(\cdot)$ denotes the trace of a matrix, and $\mathbf{1}, \mathbf{0}$ denote vectors/matrices with all entries as unity, zero respectively. Since the sample complexity of (2) in estimating (1) is prohibitively high for high-dimensional domains [29], alternative estimation methods are sought after.

### Kernel mean embeddings

This section presents a brief on the theory of kernel mean embeddings. For more details, please refer [39]. Let $k$ be a kernel defined over a domain $\mathcal{X}$ and let $\phi_k, \mathcal{H}_k$ be the kernel's canonical feature map and the canonical RKHS. Then, the kernel mean embedding of a random variable $X$ is defined as $\mu_X \equiv \mathbb{E}[\phi(X)]$. The embedding $\mu_X$ is well-defined, and $\mu_X \in \mathcal{H}_k$, whenever $k$ is normalized. Further, if (and only if) $k$ is a characteristic kernel [41], then the map $X \mapsto \mu_X$ is one-to-one. For discrete probability measures, the Kronecker delta kernel is characteristic, while for continuous measures over $\mathbb{R}^d$, the Gaussian kernel is an example of a characteristic kernel. Using these embeddings, one can compute expectations of functions of the respective random variables, whenever they exist: for e.g., $\mathbb{E}[f(X)] = \mathbb{E}[\langle f, \phi(x) \rangle]_{\mathcal{H}_k} = \langle f, \mathbb{E}[\phi(X)] \rangle_{\mathcal{H}_k} = \langle f, \mu_X \rangle_{\mathcal{H}_k} \forall f \in \mathcal{H}_k$.

The notion of kernel mean embeddings easily extends to the case of jointly defined random variables. Let $k_1, k_2$ be two kernels defined over domains $\mathcal{X}, \mathcal{Y}$ respectively. Let $\phi_1, \phi_2$ be the corresponding canonical feature maps and let $\mathcal{H}_1, \mathcal{H}_2$ be the canonical RKHSs. Then the cross-covariance operator (the joint embedding) is defined as $C_{XY} \equiv \mathbb{E}[\phi_1(X) \otimes \phi_2(Y)]$, where $\otimes$ denotes the tensor product. Again, whenever $k_1, k_2$ are individually characteristic, the map $(X, Y) \mapsto C_{XY}$ is one-to-one and $\mathbb{E}[h(X,Y)] = \langle h, C_{XY} \rangle_{\mathcal{H}_1 \otimes \mathcal{H}_2} \forall h \in \mathcal{H}_1 \otimes \mathcal{H}_2$. Analogously, one can define the auto-covariance operator $C_{XX} \equiv \mathbb{E}[\phi_1(X) \otimes \phi_1(X)]$.

The notion of embedding conditionals is also straight-forward: $\mu_{Y/x} \equiv \mathbb{E}[\phi_2(Y)/x]$. Additionally, one defines a conditional embedding operator $C_{Y/X} : \mathcal{H}_1 \mapsto \mathcal{H}_2$, such that $C_{Y/X}(\phi_1(x)) = \mu_{Y/x} \forall x \in \mathcal{X}$. For convenience of notation, $C_{Y/X}(\phi_1(x))$ is simplified as $C_{Y/X}\phi_1(x)$. With this definition, one can show that the relation $C_{Y/X}C_{XX} = C_{YX}$ holds. Also, the kernel sum rule [39] relates the conditional operator to the mean embeddings: $\mu_Y = C_{Y/X}\mu_X$.

We end with this brief with a note on the related notion of universal kernel [41]. A kernel defined over a domain $\mathcal{X}$ is universal if and only if its RKHS is dense in the set of all continuous functions over $\mathcal{X}$ [5]. If $k_1, k_2$ are universal over $\mathcal{X}, \mathcal{Y}$ respectively, then $k = k_1 k_2$ is universal over $\mathcal{X} \times \mathcal{Y}$. Moreover, $\phi_k(x, y) = \phi_1(x) \otimes \phi_2(y) \forall x \in \mathcal{X}, \ y \in \mathcal{Y}$. Hence, the RKHS $\mathcal{H}_1 \otimes \mathcal{H}_2$ is dense in the set of all continuous functions over $\mathcal{X} \times \mathcal{Y}$. Finally, universal kernels are also characteristic.

## 3 Proposed Methodology

We begin by re-formulating (1) solely in terms of kernel mean embeddings and operators. Let $k_1, k_2$ be characteristic kernels defined over $\mathcal{X}, \mathcal{Y}$ respectively. Let $\phi_1, \phi_2$ and $\mathcal{H}_1, \mathcal{H}_2$ denote the canonical feature maps and the reproducing kernel Hilbert spaces (RKHS) corresponding to the kernels $k_1, k_2$ respectively. Let $\langle \cdot, \cdot \rangle_{\mathcal{H}}, \| \cdot \|_{\mathcal{H}}$ denote the default inner-product, norm in the RKHS $\mathcal{H}$. Let $\mu_s \equiv \mathbb{E}_{X \sim p_s}[\phi_1(X)]$, $\mu_t \equiv \mathbb{E}_{Y \sim p_t}[\phi_2(Y)]$ denote the kernel mean embeddings of the marginals $p_s, p_t$ respectively. Let $\Sigma_{ss} \equiv \mathbb{E}_{X \sim p_s}[\phi_1(X) \otimes \phi_1(X)]$ and $\Sigma_{tt} \equiv \mathbb{E}_{Y \sim p_t}[\phi_2(Y) \otimes \phi_2(Y)]$ denote the auto-covariance embeddings of $p_s, p_t$ respectively. Recall that $\otimes$ denotes tensor product.

Since the variable in (1), $\pi$, is a joint measure, the cross-covariance operator, $\mathcal{U} = \mathbb{E}_{(X,Y) \sim \pi}[\phi_1(X) \otimes \phi_2(Y)]$, is the suitable kernel mean embedding to be employed. However,

since the constraints involve the marginals of $\pi$, denoted by $\pi_1, \pi_2$; it is natural to employ the kernel sum rule [39], which relates the cross-covariance operator, $\mathcal{U}$, to the marginal embeddings, $\mu_1 \equiv \mathbb{E}_{X \sim \pi_1} [\phi_1(X)], \mu_2 \equiv \mathbb{E}_{Y \sim \pi_2} [\phi_2(Y)]$, via the conditional embedding operators, $\mathcal{U}_1, \mathcal{U}_2$, and the auto-covariance operators, $\Sigma_1^{\mathcal{U}} \equiv \mathbb{E}_{X \sim \pi_1} [\phi_1(X) \otimes \phi_1(X)], \Sigma_2^{\mathcal{U}} \equiv \mathbb{E}_{Y \sim \pi_2} [\phi_2(Y) \otimes \phi_2(Y)]$. The relations between these operators and embeddings follow from the definition of conditional embedding and the kernel sum rule [39]:

$$\mathcal{U} = \Sigma_1^{\mathcal{U}} \mathcal{U}_1^{\top} = \mathcal{U}_2 \Sigma_2^{\mathcal{U}}, \; \mathcal{U}_1 \mu_1 = \mu_2, \; \mathcal{U}_2 \mu_2 = \mu_1. \tag{3}$$

Here, $M^{\top}$ denotes the adjoint of $M$.

In order to re-write the objective using the above operators, we assume that the cost function, $c(\cdot, \cdot)$, can be embedded in $\mathcal{H}_1 \otimes \mathcal{H}_2$. This assumption is trivially true if the domains are discrete. However, in case of continuous domains this need not be true, in general. Hence we additionally assume that the kernel(s) corresponding to continuous domain(s) is(are) universal and that the cost function, $C(\cdot, \cdot)$, is continuous in that(those) continuous variable(s). It then follows that $c(\cdot, \cdot)$ can be arbitrarily closely approximated by elements in $\mathcal{H}_1 \otimes \mathcal{H}_2$ [41]. Note that universal kernels are well-studied and known for non-standard domains too [5]. These very mild assumptions are summarized below:

**Assumption 1.** *Both kernels $k_1, k_2$ are characteristic. Moreover, if $k_i$ is over a continuous domain, then it is universal.*

**Assumption 2.** *We assume that $c \in \mathcal{H}_1 \otimes \mathcal{H}_2$, where $c$ denotes either the exact function or the (arbitrarily) close approximation of it that can be embedded.*

Note that the objective in (1) can be written as: $\mathbb{E}[c(X, Y)] = \langle c, \mathcal{U} \rangle_{\mathcal{H}_1 \otimes \mathcal{H}_2}$. Using this and (3), leads to the following kernel embedding formulation for OT:

$$\begin{aligned}
\min_{\mathcal{U} \in \mathcal{E}_{21}, \mathcal{U}_1 \in \mathcal{L}_{12}, \mathcal{U}_2 \in \mathcal{L}_{21}} \quad & \langle c, \mathcal{U} \rangle_{\mathcal{H}_1 \otimes \mathcal{H}_2} \\
\text{s.t.} \quad & \mathcal{U}_1 \mu_s = \mu_t, \; \mathcal{U}_2 \mu_t = \mu_s, \\
& \mathcal{U} = \Sigma_{ss} \mathcal{U}_1^{\top}, \Sigma_1^{\mathcal{U}} = \Sigma_{ss}, \; \mathcal{U} = \mathcal{U}_2 \Sigma_{tt}, \Sigma_2^{\mathcal{U}} = \Sigma_{tt},
\end{aligned} \tag{4}$$

where $\mathcal{L}_{ij}$ is the set of all linear operators from $\mathcal{H}_i \mapsto \mathcal{H}_j$, and $\mathcal{E}_{21} \equiv \{ \mathcal{U} \in \mathcal{L}_{21} \mid \exists p \in \mathcal{M}^1(\mathcal{X}, \mathcal{Y}) \ni \mathcal{U} = \mathbb{E}_{(X,Y) \sim p} [\phi_1(X) \otimes \phi_2(Y)] \}$ is the set of all valid cross-covariance operators. Note that the constraints $\mathcal{U} = \Sigma_{ss} \mathcal{U}_1^{\top}, \Sigma_1^{\mathcal{U}} = \Sigma_{ss} \Rightarrow \mathcal{U} = \Sigma_1^{\mathcal{U}} \mathcal{U}_1^{\top}$, which in turn gives that $\mathcal{U}_1$ is a valid conditional embedding associated with $\mathcal{U}$. However, we keep the former couple of constraints rather than the later one because i) there is no loss of generality ii) they will lead to an elagant representor theorem, kernelization and efficient optimization, as will be clear later. Analogous comments hold for the couple $\mathcal{U} = \mathcal{U}_2 \Sigma_{tt}, \Sigma_2^{\mathcal{U}} = \Sigma_{tt}$.

The equivalence of (4) and (1) follows from the one-to-one correspondence between the measures involved and their kernel embeddings, which is guaranteed by the characteristic kernels, and from the crucial embedding characterizing constraint: $\mathcal{U} \in \mathcal{E}_{21}$. Without this characterizing constraint, the formulation is not meaningful. We summarize the above re-formulation in the following theorem:

**Theorem 1.** *Under Assumptions 1-2, the Kantorovich formulation of OT, (1), is equivalent to (4).*

Note that unlike existing formulae for the operator embeddings [39], which eliminate two of the three operators $\mathcal{U}, \mathcal{U}_1, \mathcal{U}_2$; we critically preserve all of them in (4). This is because they facilitate efficient regularization in the statistical estimation set-up and lead to efficient algorithms (as will be shown later). Also, the characterization of embedding, $\mathcal{E}_{21}$, is included only for the cross-covariance, and not explicitly included for the conditional operators. This is fine because the conditional operators are well-defined given the cross-covariance, and the auto-covariances.

The key advantage of the proposed formulation (4) over (1) is that the sample based estimates for kernel mean embeddings of the marginals, which are known to have dimension-free sample complexities, can be employed directly in the statistical OT setting.

### 3.1 Re-formulation as Learning Embedding problem

As motivated earlier, we aim to employ the standard sample based estimates for the kernel mean embeddings of the marginals in the re-formulation (4). To this end, let the estimates for the marginal kernel mean embeddings be denoted by: $\hat{\mu}_s \equiv \frac{1}{m} \sum_{i=1}^{m} \phi_1(x_i)$ and $\hat{\mu}_t \equiv \frac{1}{n} \sum_{j=1}^{n} \phi_2(y_j)$. Likewise,

the estimates of the auto-covariance embeddings are given by $\hat{\Sigma}_{ss} \equiv \frac{1}{m} \sum_{i=1}^{m} \phi_1(x_i) \otimes \phi_1(x_i)$ and $\hat{\Sigma}_{tt} \equiv \frac{1}{n} \sum_{j=1}^{n} \phi_2(y_j) \otimes \phi_2(y_j)$.

In the statistical OT setting, the cost function, $c$, is only available at the given samples. In continuous domains, there will exist many functions in the RKHS that will exactly match $c$, when restricted to the samples. Each such choice will lead to a valid estimator. We choose $\hat{c}$ to be the orthogonal projection of $c$ onto the samples: $\hat{c} \equiv \sum_{i=1}^{m} \sum_{j=1}^{n} \rho_{ij}^* \phi_1(x_i) \otimes \phi_2(y_j)$, where $\rho^* \equiv \arg\min_\rho \left\| c - \sum_{i=1}^{m} \sum_{j=1}^{n} \rho_{ij} \phi_1(x_i) \otimes \phi_2(y_j) \right\|_{\mathcal{H}_1 \otimes \mathcal{H}_2}$ and $\|\cdot\|_{\mathcal{H}_1 \otimes \mathcal{H}_2}$ is the Hilbert-Schmidt operator norm. Straight-forward computation shows that $\rho^* = (G_1 \odot G_2)^{-1} \mathcal{C}$, where $G_1$ and $G_2$ are the gram-matrices with $k_1$ and $k_2$ over the samples $x_1, \ldots, x_m$ and $y_1, \ldots, y_n$ respectively, and $\odot$ denotes the element-wise product. Recall that $\mathcal{C} \in \mathbb{R}^{m \times n}$ denotes the cost matrix with with $(i,j)^{th}$ entry as $c(x_i, y_j)$. For universal kernels, it follows that $\hat{c}$ will be equal to $c$ at the given samples, and hence the above is a valid choice for estimation. In addition, the above choice of $\hat{c}$ helps us in proving the representer theorem (Theorem 3).

Now, employing these estimates in (4) must be performed with caution as i) the equality constraints now will be in the (potentially infinite dimensional) RKHS, ii) more importantly, matching the estimates exactly will lead to overfitting. Hence, we propose to introduce appropriate regularization by insisting that there is a close match rather than an exact match. This leads to the following kernel mean embedding learning formulation:

$$
\begin{aligned}
\min_{\mathcal{U} \in \mathcal{E}_{21}, \mathcal{U}_1 \in \mathcal{L}_{12}, \mathcal{U}_2 \in \mathcal{L}_{21}} \quad & \langle \hat{c}, \mathcal{U} \rangle_{\mathcal{H}_1 \otimes \mathcal{H}_2} \\
\text{s.t.} \quad & \|\mathcal{U}_1 \hat{\mu}_s - \hat{\mu}_t\|_{\mathcal{H}_2} \leq \Delta_1, \ \|\mathcal{U}_2 \hat{\mu}_t - \hat{\mu}_s\|_{\mathcal{H}_1} \leq \Delta_2, \\
& \left\| \mathcal{U} - \hat{\Sigma}_{ss} \mathcal{U}_1^\top \right\|_{\mathcal{H}_1 \otimes \mathcal{H}_2} \leq \vartheta_1, \left\| \mathcal{U} - \mathcal{U}_2 \hat{\Sigma}_{tt} \right\|_{\mathcal{H}_1 \otimes \mathcal{H}_2} \leq \vartheta_2, \\
& \|\Sigma_1^{\mathcal{U}} - \hat{\Sigma}_{ss}\|_{\mathcal{H}_1 \otimes \mathcal{H}_1} \leq \zeta_1, \|\Sigma_2^{\mathcal{U}} - \hat{\Sigma}_{tt}\|_{\mathcal{H}_2 \otimes \mathcal{H}_2} \leq \zeta_2,
\end{aligned}
\tag{5}
$$

where $\Delta_1, \Delta_2, \vartheta_1, \vartheta_2, \zeta_1, \zeta_2$ are regularization hyper-parameters introduced to prevent overfitting to the estimates.

## 3.2 Statistical Analysis of the Learning Formulation

The proposed embedding learning formulation (5) is an approximation to the OT problem (4) because of two reasons: i) the regularization hyper-parameters $\Delta_1, \Delta_2, \vartheta_1, \vartheta_2, \zeta_1, \zeta_2$, which are non-zero (positive) ii) sample-based estimates $(\hat{c}, \hat{\mu}_s, \hat{\mu}_t, \hat{\Sigma}_{ss}, \hat{\Sigma}_{tt})$ are employed. While the effect of the former is clear, for e.g., as the hyper-parameters $\to 0$, the approximation error, $\epsilon$, goes to zero; the sample complexity for estimation is more insightful. To this end we present the following theorem:

**Assumption 3.** *Let us assume that the kernels are normalized/bounded i.e., $\max_{x \in \mathcal{X}} k_1(x,x) = 1, \max_{y \in \mathcal{Y}} k_2(y,y) = 1$.*

**Theorem 2.** *Let $g\left(\hat{c}, \hat{\mu}_s, \hat{\mu}_t, \hat{\Sigma}_{ss}, \hat{\Sigma}_{tt}\right)$ denote the optimal objective of (5) in Tikhonov form. Under Assumptions 1-3, with high probability we have that, $\left| g\left(\hat{c}, \hat{\mu}_s, \hat{\mu}_t, \hat{\Sigma}_{ss}, \hat{\Sigma}_{tt}\right) - g\left(c, \mu_s, \mu_t, \Sigma_{ss}, \Sigma_{tt}\right) \right| \leq O\left(1/\sqrt{\min(m,n)}\right)$. The constants in the RHS of the inequality are dimension-free.*

Theorem 2 shows that embedding of an $\epsilon$-optimal transport plan can recovered by solving (5) with a sample complexity that is dimension-free. The proof of this theorem is detailed in [28]. The idea is to uniformly bound the difference between the population and sample versions of each of the terms in the objective. Interestingly, each of these difference terms can either be bounded by relevant estimation errors in embedding space or by approximation errors in the RKHS, both of which are known to be dimension-free.

Note that the regularization in (5) is based on the Maximum Mean Discrepancy (MMD) distances between the kernel embeddings. This characteristic of our estimators is in contrast with the popular entropic regularization [8], or $\phi$-divergence based regularization [25] in existing OT estimators. [40] argue that MMD and $\phi$-divergence based regularization have complementary properties. Hence both are interesting to study. While the dependence on dimensionality is adversely exponential with entropic regularization, if accurate solutions are desired [14], the proposed MMD based regularization for statistical OT leads to dimension-free estimation.

## 3.3 Representer theorem & Kernelization

Interestingly, (5) admits a finite parameterization facilitating it's efficient optimization. This important result is summarized in the representer theorem below:

**Theorem 3.** *Whenever (5) is solvable, there exists an optimal solution, $\mathcal{U}^*, \mathcal{U}_1^*, \mathcal{U}_2^*$, of (5) such that $\mathcal{U}^* = \sum_{i=1}^m \sum_{j=1}^n \alpha_{ij}\phi_1(x_i) \otimes \phi_2(y_j), \mathcal{U}_1^* = \sum_{i=1}^m \sum_{j=1}^n \beta_{ji}\phi_2(y_j) \otimes \phi_1(x_i), \mathcal{U}_2^* = \sum_{i=1}^m \sum_{j=1}^n \gamma_{ij}\phi_1(x_i) \otimes \phi_2(y_j)$. Here $\alpha \in \mathbb{R}^{m \times n}, \beta \in \mathbb{R}^{n \times m}, \gamma \in \mathbb{R}^{m \times n}$ that are an optimal solution for the kernelized and convex formulation (6) given below:*

$$
\begin{aligned}
\min_{\alpha,\gamma\in\mathbb{R}^{m\times n},\beta\in\mathbb{R}^{n\times m}} \quad & tr(\alpha\mathcal{C}^\top) \\
\text{s.t.} \quad & \tfrac{1}{m^2}\mathbf{1}^\top G_1\beta^\top G_2\beta G_1\mathbf{1} - \tfrac{2}{mn}\mathbf{1}^\top G_2\beta G_1\mathbf{1} + \tfrac{1}{n^2}\mathbf{1}^\top G_2\mathbf{1} \leq \Delta_1^2 \\
& \tfrac{1}{n^2}\mathbf{1}^\top G_2\gamma^\top G_1\gamma G_2\mathbf{1} - \tfrac{2}{mn}\mathbf{1}^\top G_1\gamma G_2\mathbf{1} + \tfrac{1}{m^2}\mathbf{1}^\top G_1\mathbf{1} \leq \Delta_2^2 \\
& \left\langle G_1\alpha - \tfrac{1}{m}G_1^2\beta^\top, \alpha G_2 - \tfrac{1}{m}G_1\beta^\top G_2 \right\rangle_F \leq \vartheta_1^2, \\
& \left\langle \alpha G_2 - \tfrac{1}{n}\gamma G_2^2, G_1\alpha - \tfrac{1}{n}G_1\gamma G_2 \right\rangle_F \leq \vartheta_2^2, \\
& \|\alpha\mathbf{1} - \tfrac{1}{m}\mathbf{1}\|_{G_1\odot G_1}^2 \leq \zeta_1^2, \|\alpha^\top\mathbf{1} - \tfrac{1}{n}\mathbf{1}\|_{G_2\odot G_2}^2 \leq \zeta_2^2 \\
& \alpha \geq 0, \mathbf{1}^\top\alpha\mathbf{1} = 1,
\end{aligned}
\tag{6}
$$

*where, $G_1$ and $G_2$ are the gram-matrices with $k_1$ and $k_2$ over $x_1, \ldots, x_m$ and $y_1, \ldots, y_n$ respectively, and $\|\mathbf{x}\|_M^2 \equiv \mathbf{x}^\top M\mathbf{x}$, is the Mahalanobis squared-norm of $\mathbf{x}$.*

The proof of this theorem is detailed in [28]. Apart from standard representer theorem-type arguments, the proof includes arguments that show that the characterizing set $\mathcal{E}_{21}$ when restricted to the linear combinations of embeddings is exactly same as the convex combinations of those. This helps us replace the membership to $\mathcal{E}_{21}$ constraint by a simplex constraint.

We note that (6) is jointly convex in the variables $\alpha, \beta$, and $\gamma$. This is because the constraints are either convex quadratic or linear and the objective is also linear. Hence obtaining estimators using (6) is computationally tractable (refer also section 3.5). It is easy to verify that (6) simplifies to the discrete OT problem (2) if both the kernels are chosen to be the Kronecker delta and all the hyper-parameters are set to zero. If one of the kernel is chosen as the Kronecker delta and the other as the Gaussian kernel, then (6) can be used for semi-discrete OT in the statistical setting. Additionally, by employing appropriate universal kernels, (6) can be used for statistical OT in non-standard domains.

We end this section with a small technical note. While the cross-covariance operator obtained by solving (6) will always be a valid one; for some hyper-parameters, which are too high, it may happen that the optimal $\beta, \gamma$ induce invalid conditional embeddings. This may make computing the transport map (7) intractable. Hence, in practice, we include additional constraints $\beta, \gamma \geq 0$.

## 3.4 Proposed Optimal Map Estimator

Once the embedding of the transport plan is obtained by solving (6), generic approaches for recovering the measure corresponding to a kernel embedding, detailed in [21, 36], can be employed to recover the corresponding transport plan. Moreover, since the recovery methods in [36] have dimension-free sample complexity, the overall sample complexity for estimating the optimal transport plan hence remains dimension-free.

We estimate the Barycentric-projection based optimal transport map, $\mathcal{T}$, at any $x \in \mathcal{X}$ as follows:

$$
\begin{aligned}
\mathcal{T}(x) \equiv \operatorname*{argmin}_{y\in\mathcal{Y}} \mathbb{E}\left[c\left(y, Y\right)/x\right] &= \operatorname*{argmin}_{y\in\mathcal{Y}} \left\langle c(y, \cdot), \mathcal{U}_1^*\phi_1(x) \right\rangle, \\
&= \operatorname*{argmin}_{y\in\mathcal{Y}} \sum_{j=1}^n \left( c(y, y_j) \sum_{j=1}^n \left( \beta_{ji}^* k_1\left(x_i, x\right) \right) \right),
\end{aligned}
\tag{7}
$$

where $\beta^*$ are obtained by solving (6) and $\mathcal{U}_1^*$ is the corresponding conditional embedding. (7) turns out to be that of finding the Karcher mean [23], whenever the cost is a squared-metric etc. Alternatively, one can directly minimize $\mathbb{E}\left[c\left(y, Y\right)/x\right]$ with respect to $y \in \mathcal{Y}$ using stochastic gradient descent (SGD). The following theorem summarizes the consistency with SGD:

**Theorem 4.** *Let the cost be a metric or it's powers greater than unity and let $\mathcal{Y}$ be compact. Then the SGD based estimator for $\mathcal{T}$ has a sample complexity that remains dimension-free.*

An advantage with our map estimator is that it can be computed even at out-of-sample $x \in \mathcal{X}$. This is possible because of the implicit smoothing induced by the kernel.

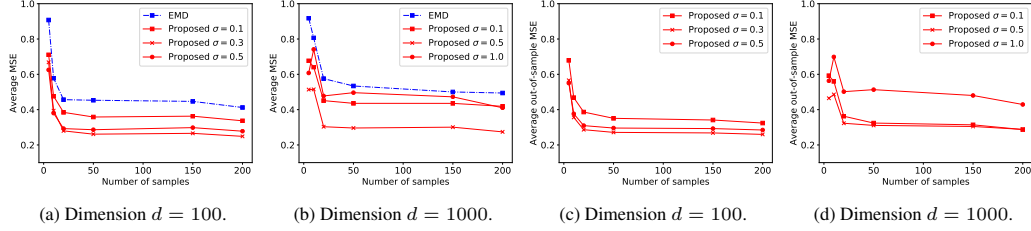

| (a) Dimension $d = 100$. | (b) Dimension $d = 1000$. | (c) Dimension $d = 100$. | (d) Dimension $d = 1000$. |

Figure 1: (a) & (b) plots the average MSE obtained by the proposed approach and the EMD (2) on the problem of learning the transport map between multivariate Gaussians; (c) & (d) plots average out-of-sample MSE obtained by the proposed approach with the learned transport map.

## 3.5   A Special Case

Though (6) can be solved using off-the-shelf convex solvers, the structure in the proposed problem can be exploited to derive efficient alternating directions method of multipliers (ADMM) based solvers. Further speed-up may be obtained in the special case when $\epsilon_i = 0$ in (6). This simplifies the constraints corresponding to $\epsilon_1$ and $\epsilon_2$ in (6) as $\alpha = (1/m)G_1\beta^\top$ and $\alpha = (1/n)\gamma G_2$, respectively. Using this and re-writing (6) in Tikhonov form leads to the following optimization problem:

$$\min_{\alpha \in \mathcal{A}_{mn}, \beta \geq \mathbf{0}, \gamma \geq \mathbf{0}} \quad tr(\alpha \mathcal{C}^\top) \quad +\lambda_1 \left\| \alpha\mathbf{1} - \tfrac{1}{m}\mathbf{1} \right\|_{G_1}^2 + \lambda_2 \left\| \alpha^\top\mathbf{1} - \tfrac{1}{n}\mathbf{1} \right\|_{G_2}^2$$
$$+\nu_1 \left\| \alpha\mathbf{1} - \tfrac{1}{m}\mathbf{1} \right\|_{G_1 \odot G_1}^2 + \nu_2 \left\| \alpha^\top\mathbf{1} - \tfrac{1}{n}\mathbf{1} \right\|_{G_2 \odot G_2}^2 \tag{8}$$
$$\text{s.t.} \qquad\qquad \alpha = \tfrac{1}{m}G_1\beta^\top, \alpha = \tfrac{1}{n}\gamma G_2,$$

where $\mathcal{A}_{mn} = \{ x \in \mathbb{R}^{m \times n} \mid x \geq \mathbf{0}, \mathbf{1}^\top x \mathbf{1} = 1 \}$ and $\lambda_i > 0, \nu_i > 0$ are the regularization hyper-parameter corresponding to $\Delta_i^2, \zeta_i^2$ in (6). The ADMM updates for (8) are discussed in [28]. Without loss of generality, if we assume $m \geq n$, the per iteration cost of ADMM algorithm for (8) is $O(m^3)$.

## 4   Related Work

A popular strategy for performing continuous statistical OT is to simply employ the sample based plug-in estimates for the marginals. This reduces the statistical OT problem to the classical discrete OT problem, for which efficient algorithms exist [8, 1]. However, the sample complexity of the discrete OT based estimation is plagued with the curse of dimensionality [29].

Many approaches for alleviating the curse of dimensionality exist. For e.g., [14] propose entropic regularization. However, their results (e.g., theorem 3 in [14]) show that the curse of dimensionality is not completely removed, especially if accurate solutions are desired. Empirical results in [11, refer Figures 4 and 5] confirm that the quality of the solution degrades very quickly with entropic regularization. [29, 12] make low-rank assumptions, which may not be satisfied in all applications. Further, as per theorem 1 in [29], the dependence on $d$ still exists. Similar comments hold for [18], which makes a smoothness assumption.

While the approach of [15] efficiently estimates the optimal dual objective, recovering the optimal transport plan from the dual's solution again requires the knowledge of the exact marginals (refer proposition 2.1 in [15]). Since estimating distributions in high-dimensional settings is known to be challenging, this alternative is not attractive for applications where the transport plan is required, e.g., domain adaptation [7],ecological inference [27], data alignment [2] etc.

On passing we note that though there are existing works that employ kernels in context of OT [15, 32, 45, 30], none of them use the notion of kernel embedding of distributions and limit the use of kernels to either function approximation or computing the MMD distance. Though relations between Wasserstein and MMD distance [11] exist, none of them explore regularization with MMD distances.

Table 1: Accuracy obtained on the target domains of the Office-Caltech dataset.

| | (a) In-sample source dataset | | | | (b) Out-of-sample source dataset | | |
|---|---|---|---|---|---|---|---|
| Task | EMD | OTLin [31] | OTKer [31] | Proposed | OTLin [31] | OTKer [31] | Proposed |
| $A \to C$ | 80.68 | 82.92 | 83.07 | **85.24** | 56.75 | 79.11 | **84.42** |
| $A \to D$ | 72.66 | 82.28 | 82.53 | **84.05** | 79.49 | 82.79 | **84.81** |
| $A \to W$ | 69.05 | **77.70** | 76.35 | 77.57 | 55.41 | 76.35 | **82.16** |
| $C \to A$ | 82.61 | 88.31 | 88.09 | **90.49** | 87.79 | 84.54 | **90.71** |
| $C \to D$ | 68.35 | **79.75** | 78.99 | 77.97 | **81.01** | 74.94 | 78.73 |
| $C \to W$ | 65.54 | 71.89 | 70.00 | **74.46** | 70.00 | 68.11 | **75.27** |
| $D \to A$ | 81.50 | 88.57 | 85.23 | **91.09** | 64.53 | 81.95 | **87.56** |
| $D \to C$ | 76.51 | 82.17 | 78.22 | **86.52** | 43.67 | 72.79 | **81.33** |
| $D \to W$ | 91.89 | **97.57** | 96.35 | 96.35 | **90.04** | 82.02 | 89.60 |
| $W \to A$ | 71.22 | 80.00 | 76.23 | **87.54** | 60.09 | 73.88 | **79.10** |
| $W \to C$ | 69.55 | 77.58 | 73.72 | **80.96** | 49.34 | 63.17 | **76.83** |
| $W \to D$ | 80.76 | **97.72** | 96.20 | 96.20 | **95.95** | 90.89 | 93.16 |
| Average | 75.86 | 83.87 | 82.08 | **85.70** | 69.51 | 77.54 | **83.64** |

## 5 Experiments

We evaluate our estimator for the transport map (7) on both synthetic and real-world datasets. Our code is available at `https://www.iith.ac.in/~saketha/research.html`. Additional details on the experiments are available in the technical report [28].

### 5.1 Learning OT map between multivariate Gaussian distributions

The optimal transport map between two Gaussian distributions $g_{source} = N(m_1, \Sigma_1)$ and $g_{target} = N(m_2, \Sigma_2)$ with squared Euclidean cost has a closed form expression [33] given by $T : x \mapsto m_2 + A(x - m_1)$, where $A = \Sigma_1^{-\frac{1}{2}}(\Sigma_1^{\frac{1}{2}}\Sigma_2\Sigma_1^{\frac{1}{2}})^{\frac{1}{2}}\Sigma_1^{-\frac{1}{2}}$. We compare the proposed estimator (7) in terms of the deviation from the optimal transport map.

**Experimental setup**: We consider mean zero Gaussian distributions with unit-trace covariances and sample equal number ($m$) of source and target data points, where $m \in \{10, 20, 50, 100, 150, 200\}$ and $d \in \{100, 1000\}$. The covariance matrices are computed as $\Sigma_1 = V_1 V_1^\top / \|V_1\|_F$ and $\Sigma_2 = V_2 V_2^\top / \|V_2\|_F$, where $V_1 \in \mathbb{R}^{d \times d}$ and $V_2 \in \mathbb{R}^{d \times d}$ are generated randomly from the uniform distribution. Our approach employs the Gaussian kernels, $k(x, z) = \exp(-\|x - z\|^2 / 2\sigma^2)$. Initial experiments indicate that suitable values of $\sigma$ include those that does not yield high condition number of the Gram matrices. As a baseline, we also report the results obtained from the discrete OT estimator, henceforth referred to as EMD, learned via the discrete OT problem (2). For each $(d, m)$, we randomly sample five sets of data points and report the average mean square error (MSE).

**Results**: The results are reported in Figures 1(a) & (b). We observe that the proposed estimator obtains lower average MSE than EMD across different number of samples and dimensions. The advantage of the proposed estimator over EMD is more pronounced at higher dimension.

**Out-of-sample evaluation**: We also evaluate our estimator's ability to map out-of-sample data by sampling additional $m_{oos} = 200$ points from the source distributions in the above experiments. These source points are not used to learn the estimator. The results are reported in Figure 1(c) & (d). We observe that the performance on out-of-sample data points are similar to the in-sample data points (Figures 1(a) & (b)). In addition, the average out-of-sample MSE generally decreases with increasing number of samples since a better estimator is learned with more number of samples. Overall, the results illustrate the utility of the proposed approach for out-of-sample estimation. It should be noted that the baseline EMD cannot map out-of-sample data points.

### 5.2 Domain adaptation

We experiment on the Caltech-Office dataset [17], which contains images from four domains: Amazon (online retail), the Caltech image dataset, DSLR (images taken from a high resolution DSLR

camera), and Webcam (images taken from a webcam). The domains vary with respect to factors such as background, lightning conditions, noise, etc. The number of examples in each domain is: 958 (Amazon), 1123 (Caltech), 157 (DSLR), and 295 (Webcam). Each domain has images from ten classes and in turn is considered as the source or the target domain. We perform multi-class classification in the target domain given the labeled data only from the source domain. Using OT, we first transport the labeled source domain data-points to the target domain and then learn a classifier for the target domain using the adapted source data-points. Overall, there are twelve adaptation tasks (e.g., task $A \rightarrow C$ has Amazon as the source and Caltech as the target domain). We employ DeCAF6 features to represent the images [10, 31, 6].

**Experimental setup**: For learning transport plan, we randomly select ten images per class for the source domain (eight per class when DSLR is the source, due to its sample size). The remaining samples of the source domain is marked as out-of-sample source data-points. The target domain is partitioned equally into training and test sets. The transport map is learned using the source-target training sets. The 'in-sample' accuracy is then evaluated on the target's test set. We also evaluate the quality of our out-of-sample estimation as follows. Instead of projecting the source training set samples onto the target domain, we project only the out-of-sample (OOS) source data-points and compute the accuracy over the target's test set. It should be noted that the transport model has not been learned on the OOS data-points, such mappings may not be as accurate as the in-sample mapping. The OOS evaluation assesses the downstream effectiveness of OOS estimation on domain adaptation. Out-of-sample estimation is especially attractive in big data and online applications. The classification in the target domain is performed using a 1-Nearest Neighbor classifier [17, 31, 6]. The above experimentation is performed five times and the average in-sample and out-of-sample accuracy are reported in Tables 1(a) & 1(b), respectively.

**Baselines**: We compare our approach with EMD, OTLin [31], and OTKer [31]. Both OTLin and OTKer aim to solve the discrete optimal transport problem and also learn a transformation approximating the corresponding transport map in a joint optimization framework. OTLin learns a linear transformation while OTKer learns a non-linear transformation (e.g., via Gaussian kernel). The learned transformation allows OTLin and OTKer to perform out-of-sample estimation as well. We use the Python Optimal Transport (POT) library (`https://github.com/PythonOT/POT`) implementations of OTLin and OTKer in our experiments.

**Results**: We observe from Tables 1(a) & 1(b) that the proposed approach outperforms the baselines, obtaining the best in-sample and out-of-sample (OOS) accuracy. As discussed, the in-sample accuracy is likely to be better than out-of-sample accuracy (for any approach). Interestingly, for a few tasks with Amazon and Caltech as the source domains, the OOS accuracy of our approach is comparable to our in-sample accuracy. In these domains, the OOS set is larger than the training set. The proposed OOS estimation is able to exploit this and provide an effective knowledge transfer. Conversely, we observe a drop in our OOS accuracy (when compared with the corresponding in-sample accuracy) in tasks with DSLR and Webcam as the source domains since the size of OOS set is quite small and hence lesser potential for knowledge transfer. On the other hand, OTLin suffers a significant drop in OOS performance, likely due the the overfitting of the learned linear transformation on the source training points. While OTKer has better OOS performance than OTLin, it has more variance between in-sample and out-of-sample performance than the proposed approach.

# 6 Conclusions

The idea of employing kernel embeddings of distributions in OT seems promising, especially in the continuous case. It not only leads to sample complexities that are dimension-free, but also provides a new regularization scheme based on MMD distances, which is complementary to existing $\phi$-divergence based regularization.

While the optimal solution of the proposed MMD regularized formulation recovers the transport plan, the objective value does not seem to have any special use. On the contrary, it has been shown that with entropic, $\phi$-divergence based regularizations the optimal objectives lead to notions of Sinkhorn divergences [11] and Hillinger-Kantorovich metrics [25]. We make an initial observation that in the special case in section 3.5, the objective in (8), resembles that defining the Hillinger-Kantorovich metrics very closely. Hence, we conjecture that our optimal objective in this special case may also define a new family of metrics. However, we postpone such connections (if any) to future work.

## Acknowledgements

JSN would like to thank Manohar Kaul and Jatin Chauhan for initial discussions. Part of this work was done while JSN was visiting Microsoft IDC, Hyderabad, and JSN thanks Microsoft for the visiting opportunity. PJ would like to thank Bamdev Mishra for useful discussions.

## Broader Impact Statement

Our work complements the ongoing research explorations towards a better understanding of sample complexity in regularized optimal transport problems. Our contributions are mainly theoretical with some empirical results on standard optimal transport applications. Overall, this work does not present any foreseeable societal consequence.

## Footnotes

[1]Here we use a simplified notation for $O(m^{-1/2})$, where the terms involving $\epsilon$ are ignored. Nevertheless, all the terms/constants ignored are indeed independent of the dimension (and $m$).

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
