[Supplementary Material 1]

# Supplementary material: Statistical Optimal Transport posed as Learning Kernel Mean Embedding

## Abstract

The objective in statistical Optimal Transport (OT) is to consistently estimate the optimal transport plan/map solely using samples from the given source and target marginal distributions. This work takes the novel approach of posing statistical OT as that of learning the transport plan's kernel mean embedding from sample based estimates of marginal embeddings. A key result is that, under mild conditions, the sample complexity of the resulting estimator for the optimal transport plan as well as that for the Barycentric-projection based optimal transport map are dimension-free. Moreover, the implicit smoothing in the kernel embeddings not only improves the quality of finite sample estimation but also enables out-of-sample estimation. Also, complementary to existing $\phi$-divergence (entropy) based regularization techniques, our estimator employs a maximum mean discrepancy (MMD) based regularization to avoid over-fitting the samples. We present an appropriate representer theorem that leads to a kernelized convex formulation, which can then be potentially used to perform OT even in non-standard domains. Empirical results illustrate the efficacy of the proposed approach.

## 1 Introduction

Optimal Transport is proving to be an increasingly successful tool in solving diverse machine learning problems. Recent research shows that variants of Optimal Transport (OT) achieve state-of-the-art performance in various machine learning (ML) applications such as domain adaptation [9], NLP [2, 40, 41], robust learning [6], etc. It is also shown that OT based (Wasserstein) metrics serve as good loss functions in both supervised [15] and unsupervised [19] learning. [30] is a comprehensive monologue on the subject with focus on recent developments related to machine learning.

Given two marginal distributions over source and target domains, and a cost function between elements of the domains, the classical OT problem (Kantorovich's formulation) is that of finding the joint distribution whose marginals are equal to the given marginals, and which minimizes the expected cost with respect to this joint distribution [22]. This joint distribution is known as the (optimal) transport plan or the optimal coupling. A related object of interest for ML applications is the so-called Barycentric-projection based transport map corresponding to a transport plan (e.g., refer (11) in [34]). Though OT techniques already improve state-of-the-art in many ML applications, there are two main bottlenecks that seem to limit OT's success in ML settings:

- while continuous distributions are ubiquitous, algorithms for finding the transport plan/map over continuous domains are very scarce [18]. The situation is worse in case of non-standard domains, which are not uncommon in ML.

- the marginal distributions are never available, and merely samples from them are given. The variant of OT where the transport plan/map needs to be estimated merely using samples from the marginals is known as the statistical OT problem. In case of statistical OT over continuous domains, [18, 17, 14, 5] note that estimators that are free from the curse of dimensionality are not well-studied.

The concluding report from a recent workshop on OT (refer section 2 in [5]) summarizes that one of the major open problems in this area is to design estimators in context of continuous statistical OT whose sample complexity is not a strong function of the dimension (ideally dimension-free).

Our work focuses on this challenging and important problem of statistical OT over continuous domains, and seeks consistent estimators whose sample complexity is dimension-free. To this end, we take the novel approach of equivalently re-formulating the statistical OT problem solely in terms of the relevant kernel mean embeddings [25]. More specifically, our formulation finds the (characterizing) kernel mean embedding of a joint distribution with least expected cost, and whose marginal embeddings are close to the given-sample based estimates of the marginal embeddings. There are several advantages of this new approach:

1. because the samples based estimates of the kernel mean embeddings of the marginals are known to have sample complexities that are dimension-free, it is expected that the sample complexity remains dimension-free even for the proposed estimator of the transport plan embedding.
2. kernel embeddings provide implicit smoothness, as controlled by the kernel. Appropriate smoothness not only improves the quality of estimation, but also enable out-of-sample estimation.
3. while existing estimators employ $\phi$-divergence (or entropy) based regularization, our formulation employs Maximum Mean Discrepency (MMD) based regularization to avoid overfitting the samples. This is facilitated as MMD is the natural notion of distance in the kernel mean embedding space. As discussed in [38], MMD and $\phi$-divergence based regularization exhibit complementary properties and hence both are interesting to study.

To the best of our knowledge, existing works have not employed kernel mean embeddings explicitly in the context of OT.

A key result from this work is that, under mild conditions, the proposed estimator for the optimal transport plan as well as the (Barycentric-projection based) optimal transport map is statistically consistent with a sample complexity that remains dimension-free. Another key contribution is an appropriate representer theorem that guarantees finite characterization for the transport plan embedding, which leads to a fully kernelized and convex formulation. Thus the same formulation can potentially be used for obtaining estimators in all variants of OT: continuous, semi-discrete, and discrete, merely by switching the kernel between the Kronecker delta and the Gaussian kernels. More importantly, the same can be used to solve OT problems in non-standard domains using appropriate universal kernels [7]. Finally, we present an alternating direction method of multipliers (ADMM) based algorithm for efficiently solving the proposed formulation. Empirical results on synthetic and real-world datasets illustrate the efficacy of the proposed approach.

## 2 Background on Optimal Transport

Let $\mathcal{X}, \mathcal{Y}$ be any two sets that form locally compact Hausdorff topological spaces. We denote the set of all Radon probability measures over $\mathcal{X}$ by $\mathcal{M}^1(\mathcal{X})$; whereas we denote the set of strictly positive measures by $\mathcal{M}^1_+(\mathcal{X})$. Let $c : \mathcal{X} \times \mathcal{Y}$ denote a function that evaluates the cost between elements in $\mathcal{X}, \mathcal{Y}$ and let $p_s \in \mathcal{M}^1_+(\mathcal{X}), p_t \in \mathcal{M}^1_+(\mathcal{Y})$. Then, the Kantorovich's OT formulation [22] is:

$$\begin{aligned}
\min_{\pi \in \mathcal{M}^1(\mathcal{X}, \mathcal{Y})} \quad & \int c(x,y) \, \mathrm{d}\pi(x,y), \\
\text{s.t.} \quad & \pi_{\mathcal{X}} = p_s, \pi_{\mathcal{Y}} = p_t,
\end{aligned} \tag{1}$$

where $\pi_{\mathcal{X}}, \pi_{\mathcal{Y}}$ denote the marginal measures of $\pi$ over $\mathcal{X}, \mathcal{Y}$ respectively. An optimal solution of (1) is referred to as an optimal transport plan or optimal coupling.

**Statistical OT**: In the setting of statistical OT, the marginals $p_s, p_t$ are not available; however, iid samples from them are given. Let $\mathcal{D}_x = \{x_1, \ldots, x_m\}$ denote the set of $m$ iid samples from $p_s$ and let $\mathcal{D}_y = \{y_1, \ldots, y_n\}$ denote $n$ iid samples from $p_t$. The cost function is known only at the sample data points. Let $\mathcal{C} \in \mathbb{R}^{m \times n}$ denote the cost matrix with with $(i,j)^{th}$ entry as $c(x_i, y_j)$.

A popular way to estimate the optimal plan in (1) is to simply employ the sample based plug-in estimates for the marginals: $\hat{p}_s \equiv \frac{1}{m} \sum_{i=1}^{m} \delta_{x_i}$ and $\hat{p}_t \equiv \frac{1}{n} \sum_{j=1}^{n} \delta_{y_j}$, in place of the true (unknown) marginals. Here, $\delta$ denotes the Dirac delta function. In such a case, (1) simplifies as the standard discrete OT problem:

$$\begin{aligned} \min_{\pi \in \mathbb{R}^{m \times n}} \quad & tr(\pi \mathcal{C}^\top), \\ \text{s.t.} \quad & \pi \mathbf{1} = \tfrac{1}{m}\mathbf{1}, \pi^\top \mathbf{1} = \tfrac{1}{n}\mathbf{1}, \pi \geq \mathbf{0}, \end{aligned} \tag{2}$$

where $tr(M)$ is the trace of matrix $M$, and $\mathbf{1}, \mathbf{0}$ denote vectors/matrices with all entries as unity, zero respectively (of appropriate dimension). Since the sample complexity of (2) in estimating (1) is prohibitively high for high-dimensional domains [12], alternative estimation methods are sought after.

## 3 Proposed Methodology

We begin by re-formulating (1) solely in terms of kernel mean embeddings and operators. Let $k_1, k_2$ be characteristic kernels [16, 39] defined over $\mathcal{X}, \mathcal{Y}$ respectively. By definition, the key advantage of a characteristic kernel is that the mapping between kernel mean embeddings and $\mathcal{M}^1$ becomes one-to-one (injective). For discrete probability measures, the Kronecker delta kernel is characteristic, while for continuous measures, the Gaussian kernel is an example of a characteristic as well as a universal kernel. Let $\phi_1, \phi_2$ and $\mathcal{H}_1, \mathcal{H}_2$ denote the canonical feature maps and the reproducing kernel Hilbert spaces (RKHS) corresponding to the kernels $k_1, k_2$ respectively. Let $\langle \cdot, \cdot \rangle_{\mathcal{H}_i}, \|\cdot\|_{\mathcal{H}_i}$ denote the default inner-product, norm in the RKHS $\mathcal{H}_i$. Let $\mu_s \equiv \mathbb{E}_{X \sim p_s}[\phi_1(X)]$, $\mu_t \equiv \mathbb{E}_{Y \sim p_t}[\phi_2(Y)]$ denote the kernel mean embeddings of the marginals $p_s, p_t$ respectively. Let $\Sigma_{ss} \equiv \mathbb{E}_{X \sim p_s}[\phi_1(X) \otimes \phi_1(X)]$ and $\Sigma_{tt} \equiv \mathbb{E}_{Y \sim p_t}[\phi_2(Y) \otimes \phi_2(Y)]$ denote the auto-covariance embeddings of $p_s, p_t$ respectively. Here $\otimes$ denotes tensor product. Using these embeddings one can compute expectations of functions of the respective random variables: for e.g., $\mathbb{E}[f(X)] = \mathbb{E}[\langle f, \phi(x) \rangle] = \langle f, \mathbb{E}[\phi(X)] \rangle$ etc.

Since the variable, $\pi$, is a joint measure, the cross-covariance operator, $\mathcal{U}^\pi = \mathbb{E}_{(X,Y) \sim \pi}[\phi_1(X) \otimes \phi_2(Y)]$, is the suitable kernel mean embedding to be employed. However, the constraints involve the marginals of $\pi$, whose embeddings cannot be retrieved from the cross-covariance operator alone. Hence we also employ the conditional embedding operators, $\mathcal{U}_1^\pi, \mathcal{U}_2^\pi$, which embed the conditionals $\pi_{\mathcal{Y}/\mathcal{X}}(\cdot/\cdot)$ and $\pi_{\mathcal{X}/\mathcal{Y}}(\cdot/\cdot)$ respectively. The relations between these operators and embeddings follow from the definition of conditional embedding and the kernel sum rule [37]: $\mathcal{U} = \Sigma_{ss}\mathcal{U}_1^\top = \mathcal{U}_2\Sigma_{tt}$, $\mathcal{U}_1\mu_s = \mu_t$, $\mathcal{U}_2\mu_t = \mu_s$.

In order to re-write the objective using the above operators, we assume that the cost function, $c$, can be embedded in $\mathcal{H}_2 \otimes \mathcal{H}_1$. This assumption is trivially true if the domains are discrete. However, in case of continuous domains this need not be true, in general. Hence we additionally assume that the kernel corresponding to a continuous domain is universal and that the cost function is continuous in that continuous variable. It then follows from the definition of universal kernels that a continuous function like $c(\cdot, \cdot)$ can be arbitrarily closely approximated by elements in $\mathcal{H}_2 \otimes \mathcal{H}_1$ [39]. Note that universal kernels are well-studied and known for non-standard domains too [7].

Now, the objective in (1) can be written as: $\mathbb{E}[c(X, Y)] = \langle c, \mathcal{U} \rangle_{\mathcal{H}_2 \otimes \mathcal{H}_1}$. This leads to the following kernel embedding formulation for OT:

$$\begin{aligned} \min_{\mathcal{U}, \mathcal{U}_1, \mathcal{U}_2} \quad & \langle c, \mathcal{U} \rangle_{\mathcal{H}_2 \otimes \mathcal{H}_1} \\ \text{s.t.} \quad & \mathcal{U}_1\mu_s = \mu_t, \ \mathcal{U}_2\mu_t = \mu_s, \ \mathcal{U} = \Sigma_{ss}\mathcal{U}_1^\top = \mathcal{U}_2\Sigma_{tt}, \\ & \mathcal{U} \in \mathcal{E}(\mathcal{H}_2, \mathcal{H}_1), \mathcal{U}_1 \in \mathcal{L}(\mathcal{H}_1, \mathcal{H}_2), \mathcal{U}_2 \in \mathcal{L}(\mathcal{H}_2, \mathcal{H}_1), \end{aligned} \tag{3}$$

where $\mathcal{L}(\mathcal{H}_1, \mathcal{H}_2)$ is the set of all linear operators from $\mathcal{H}_1 \mapsto \mathcal{H}_2$, and $\mathcal{E}(\mathcal{H}_2, \mathcal{H}_1) \equiv \{\mathcal{U} \in \mathcal{L}(\mathcal{H}_2, \mathcal{H}_1) \mid \exists p \in \mathcal{M}^1(\mathcal{X}, \mathcal{Y}) \ni \mathcal{U} = \mathbb{E}_{(X,Y) \sim p}[\phi_1(X) \otimes \phi_2(Y)]\}$ is the set of all valid cross-covariance operators. The equivalence of (3) and (1) follows from the one-to-one correspondence between the measures involved and their kernel embeddings, which is guaranteed by the characteristic kernels, and from the crucial embedding characterizing constraint: $\mathcal{U} \in \mathcal{E}(\mathcal{H}_2, \mathcal{H}_1)$. Without this characterizing constraint, the formulation is not meaningful. We summarize the above re-formulation in the following theorem:

**Assumption 1.** *Both kernels $k_1, k_2$ are characteristic. Moreover, if $k_i$ is over a continuous domain, then it is universal.*

129 **Assumption 2.** *We assume that $c \in \mathcal{H}_2 \otimes \mathcal{H}_1$, where $c$ denotes either the exact function or the*
130 *(arbitrarily) close approximation of it that can be embedded.*

131 **Theorem 1.** *Under Assumptions 1-2, the Kantorovich formulation of OT (1) is equivalent to (3).*

132 Note that unlike existing formulae for the operator embeddings [37], which eliminate two of the
133 three operators $\mathcal{U}, \mathcal{U}_1, \mathcal{U}_2$; we critically preserve all of them in (3). This is because they facilitate
134 efficient regularization in the statistical estimation set-up and lead to efficient algorithms (as will be
135 shown later). Also, the characterization of embedding, $\mathcal{E}(\mathcal{H}_1, \mathcal{H}_2)$, is included only for the cross-
136 covariance, and not explicitly included for the conditional operators. This is because the conditionals
137 are well-defined given the cross-covariance, auto-covariance and marginal embeddings.

138 The main advantage of the proposed formulation (3) over (1) is that the sample based estimates
139 for kernel mean embeddings of the marginals, which are known to have dimension-free sample
140 complexities, can be employed directly in the statistical OT setting.

## 3.1 Proposed formulation for statistical OT

142 As motivated earlier, we aim to employ the standard sample based estimates for the kernel mean
143 embeddings of the marginals in the re-formulation (3). To this end, let the estimates for the marginal
144 kernel mean embeddings be denoted by: $\hat{\mu}_s \equiv \frac{1}{m} \sum_{i=1}^{m} \phi_1(x_i)$ and $\hat{\mu}_t \equiv \frac{1}{n} \sum_{j=1}^{n} \phi_2(y_j)$. Likewise,
145 the estimates of the auto-covariance embeddings are given by $\hat{\Sigma}_{ss} \equiv \frac{1}{m} \sum_{i=1}^{m} \phi_1(x_i) \otimes \phi_1(x_i)$ and
146 $\hat{\Sigma}_{tt} \equiv \frac{1}{n} \sum_{j=1}^{n} \phi_2(y_j) \otimes \phi_2(y_j)$.

147 In the statistical OT setting, the cost function, $c$, is only available at the given samples. In continuous
148 domains, there will exist many functions in the RKHS that will exactly match $c$ restricted to the
149 samples. Each such choice will lead to a valid estimator. We choose $\hat{c} \equiv \sum_{i=1}^{m} \sum_{j=1}^{n} \rho_{ij}^* \phi_1(x_i) \otimes$
150 $\phi_2(y_j)$, where $\rho^* \equiv \arg\min_\rho \left\| c - \sum_{i=1}^{m} \sum_{j=1}^{n} \rho_{ij} \phi_1(x_i) \otimes \phi_2(y_j) \right\|_{\mathcal{H}_2 \otimes \mathcal{H}_1}$ and $\| \cdot \|_{\mathcal{H}_2 \otimes \mathcal{H}_1}$ is the
151 Hilbert-Schmidt operator norm. For universal kernels, it follows that $\hat{c}$ will be equal to $c$ at the given
152 samples, and hence the above is a valid choice for estimation. In addition, the above choice of $\hat{c}$ helps
153 us in proving the representer theorem (Theorem 3).

154 Now, employing these estimates in (3) must be performed with caution as i) the equality constraints
155 now will be in the (potentially infinite dimensional) RKHS, ii) more importantly, matching the
156 estimates exactly will lead to over-fitting. Hence, we propose to introduce appropriate regularization
157 by insisting that there is a close match rather than an exact match. This leads to the following kernel
158 embedding learning formulation:

$$
\begin{aligned}
\min_{\mathcal{U}, \mathcal{U}_1, \mathcal{U}_2} \quad & \langle \hat{c}, \mathcal{U} \rangle_{\mathcal{H}_2 \otimes \mathcal{H}_1} \\
\text{s.t.} \quad & \|\mathcal{U}_1 \hat{\mu}_s - \hat{\mu}_t\|_{\mathcal{H}_2} \leq \Delta_1, \; \|\mathcal{U}_2 \hat{\mu}_t - \hat{\mu}_s\|_{\mathcal{H}_1} \leq \Delta_2, \\
& \left\| \mathcal{U} - \hat{\Sigma}_{ss} \mathcal{U}_1^\top \right\|_{\mathcal{H}_2 \otimes \mathcal{H}_1} \leq \epsilon_1, \left\| \mathcal{U} - \mathcal{U}_2 \hat{\Sigma}_{tt} \right\|_{\mathcal{H}_2 \otimes \mathcal{H}_1} \leq \epsilon_2, \\
& \mathcal{U} \in \mathcal{E}(\mathcal{H}_2, \mathcal{H}_1), \mathcal{U}_1 \in \mathcal{L}(\mathcal{H}_1, \mathcal{H}_2), \mathcal{U}_2 \in \mathcal{L}(\mathcal{H}_2, \mathcal{H}_1),
\end{aligned}
\tag{4}
$$

159 where $\Delta_1, \Delta_2, \epsilon_1, \epsilon_2$ are regularization hyper-parameters introduced to prevent overfitting to the
160 estimates. Setting $\Delta_i = 0 = \epsilon_i$ recovers the case where estimates of marginal mean embed-
161 dings and auto-covariances are exactly matched but it may lead to overfitting. Also, $\mathcal{U}_1, \mathcal{U}_2$
162 are guaranteed to be valid conditional embeddings only as $\Delta_i, \epsilon_i \to 0$. Hence, we suggest
163 $\Delta_i, \epsilon_i = O\left(1/\sqrt{\min(m,n)}\right)$, following known sample complexities for the marginal embedding
164 estimates, which are $O(1/\sqrt{m}), O(1/\sqrt{n})$ respectively [36]. Since the kernel embedding estimates
165 have sample complexities that are independent of dimension, it is expected that the statistical esti-
166 mation error with the proposed formulation (4) is also independent of dimensionality. In the next
167 theorem, we formalize the above statement:

168 **Assumption 3.** *Let us assume that the kernels are normalized/bounded i.e., $\max_{x \in \mathcal{X}} k_1(x, x) =$*
169 $1, \max_{y \in \mathcal{Y}} k_2(y, y) = 1$.

170 **Theorem 2.** *Let* $g\left(\hat{c}, \hat{\mu}_s, \hat{\mu}_t, \hat{\Sigma}_{ss}, \hat{\Sigma}_{tt}\right)$ *denote the optimal objective of (4) in*
171 *Tikhonov form. Under Assumptions 1-3, with high probability we have that,*
172 $\left| g\left(\hat{c}, \hat{\mu}_s, \hat{\mu}_t, \hat{\Sigma}_{ss}, \hat{\Sigma}_{tt}\right) - g\left(c, \mu_s, \mu_t, \Sigma_{ss}, \Sigma_{tt}\right) \right| \leq O\left(1/\sqrt{\min(m,n)}\right)$. *The constants in*
173 *the RHS of the inequality are dimension-free.*

174 Theorem 2 shows that with appropriate regularization one can obtain statistically consistent estimators
175 for the embedding of the optimal transport plan by solving (4). More importantly, its proves that

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

$$
\begin{aligned}
&\min_{\alpha \in \mathcal{A}_{mn}, \beta \in \mathbb{R}^{m \times n} \geq 0, \gamma \in \mathbb{R}^{m \times n} \geq 0} && tr(\alpha \mathcal{C}^\top) + \lambda_1 \left\| \alpha \mathbf{1} - \tfrac{1}{m} \mathbf{1} \right\|_{G_1}^2 + \lambda_2 \left\| \alpha^\top \mathbf{1} - \tfrac{1}{n} \mathbf{1} \right\|_{G_2}^2 \\
&\text{s.t.} && \alpha = \tfrac{1}{m} G_1 \beta^\top, \alpha = \tfrac{1}{n} \gamma G_2
\end{aligned} \tag{7}
$$

where $\mathcal{A}_{mn} = \{x \in \mathbb{R}^{m \times n} \mid x \geq \mathbf{0}, \mathbf{1}^\top x \mathbf{1} = 1\}$ and $\lambda_i > 0$ are the regularization hyper-parameter corresponding to $\Delta_i^2$ in (5). The updates for the ADMM are summarized below:

$$
\begin{aligned}
\alpha^{(k+1)} &:= \underset{\alpha \in \mathcal{A}_{mn}}{\operatorname{argmin}} \rho \left\| \alpha + \tfrac{1}{2}\left( D_1^{(k)} + D_2^{(k)} + \tfrac{\mathcal{C}}{\rho} - \tfrac{\gamma^{(k)} G_2}{n} - \tfrac{G_1 \beta^{(k)\top}}{m} \right) \right\|^2 \\
&\qquad + \lambda_1 \left\| \alpha \mathbf{1} - \tfrac{1}{m} \mathbf{1} \right\|_{G_1}^2 + \lambda_2 \left\| \alpha^\top \mathbf{1} - \tfrac{1}{n} \mathbf{1} \right\|_{G_2}^2, \\
\beta^{(k+1)} &:= \underset{\beta \geq 0}{\operatorname{argmin}} \left\| \alpha^{(k+1)} + D_1^{(k)} - \tfrac{G_1 \beta^\top}{m} \right\|^2, \\
\gamma^{(k+1)} &:= \underset{\gamma \geq 0}{\operatorname{argmin}} \left\| \alpha^{(k+1)} + D_2^{(k)} - \tfrac{\gamma G_2}{n} \right\|^2, \\
D_1^{(k+1)} &:= D_1^{(k)} + \left( \alpha^{(k+1)} - \tfrac{G_1 \beta^{(k+1)\top}}{m} \right), \\
D_2^{(k+1)} &:= D_2^{(k)} + \left( \alpha^{(k+1)} - \tfrac{\gamma^{(k+1)} G_2}{n} \right),
\end{aligned}
$$

where $D_1$ and $D_2$ are the dual variables corresponding to the constraints $\alpha = (1/m)G_1\beta^\top$ and $\alpha = (1/n)\gamma G_2$ in (7), respectively. The optimization problems with respect to $\alpha, \beta$, and $\gamma$ can be solved efficiently using popular algorithms like conditional gradient descent, mirror descent, co-ordinate descent, conjugate gradients, etc. Since the convergence rate of these algorithms is either independent or almost independent (logarithmically dependent) on the dimensionality of the problem, the computational cost (after neglecting log factors, if any) of solving for: $\alpha$ is $O(mn)$, $\beta$ is $O(m^2n)$, and $\gamma$ is $O(mn^2)$. The updates for $D_1$ and $D_2$ have computational costs: $O(m^2n)$ and $O(mn^2)$. Without loss of generality, if we assume $m \geq n$, the per iteration cost of ADMM is $O(m^3)$.

As noted earlier, in typical cases where the hyper-parameters $\Delta_i$ are small enough, explicit constraints $\beta \geq 0, \gamma \geq 0$ are not needed:

$$
\min_{\alpha \in \mathcal{A}_{mn}} \quad tr(\alpha \mathcal{C}^\top) + \lambda_1 \left\| \alpha \mathbf{1} - \tfrac{1}{m} \mathbf{1} \right\|_{G_1}^2 + \lambda_2 \left\| \alpha^\top \mathbf{1} - \tfrac{1}{n} \mathbf{1} \right\|_{G_2}^2 \tag{8}
$$

Hence the computational cost in this special case is $O(mn)$, which is linear, and hence comparable to that of Sinkhorn algorithm popularly used to solve the discrete OT problem.

Figure 1: Performance on the proposed estimator for the transport map (6) and the discrete OT estimator, EMD, on the problem of learning the optimal transport map between two multivariate Gaussian distributions. We observe that the proposed estimator outperforms EMD, especially in higher dimensions.

## 4 Related Work

A popular strategy for performing continuous statistical OT is to simply employ the sample based plug-in estimates for the marginals. This reduces the statistical OT problem to the classical discrete OT problem, for which efficient algorithms exist [10, 1]. However, the sample complexity of the discrete OT based estimation is plagued with the curse of dimensionality [12]. [18, 17, 14] note that estimators that are free from the curse of dimensionality are not well-studied and propose alternatives.

While the approach of [18] efficiently estimates the optimal dual objective, recovering the optimal transport plan from the dual's solution again requires the knowledge of the exact marginals (refer proposition 2.1 in [18]). Since estimating distributions in high-dimensional settings is known to be challenging, this alternative is not attractive for applications where the transport plan is required, e.g., domain adaptation [9] and ecological inference [26], etc.

[17] observe that continuous statistical OT is the major bottleneck for applying OT in ML problems and propose an entropic regularization based alternative. However, their results (e.g., theorem 3 in [17]) show that the curse of dimensionality is not completely removed, especially if accurate solutions are desired. Empirical results in [13, refer Figures 4 and 5] confirm that the quality of the solution degrades very quickly with entropic regularization. The alternative in [14] makes strong low-rank based assumptions, which may not be realistic in all applications. Infact, the report on a recent workshop on OT (refer section 2 in [5]) summarizes that one of the major open problems in this area is to design estimators for continuous statistical OT whose sample complexity is not a strong function of dimensionality (ideally dimension-free).

On passing we note that though there are existing works that employ kernels in context of OT [18, 29, 42, 27], none of them use the notion of kernel embedding of distributions and limit the use of kernels to either function approximation or computing MMD distance. Though relations betweeen Wasserstein and MMD distance [13] exist, none of them explore regularization with MMD distances.

## 5 Experiments

We evaluate our estimator for the transport map (6) on both synthetic and real-world datasets.

(a) Dimension $d = 5$.  (b) Dimension $d = 10$.  (c) Dimension $d = 50$.

(d) Dimension $d = 100$.  (e) Dimension $d = 500$.  (f) Dimension $d = 1000$.

Figure 2: Average out-of-sample mean square error (MSE) obtained by the proposed approach on the problem of learning the optimal transport map between two multivariate Gaussian distributions. In general, the average out-of-sample MSE decrease with increasing number of data points sampled to learn the estimator (the x-axis). We also observe that the best results obtained by proposed solution is robust to the dimensionality of the data points.

## 5.1 Learning OT map between multivariate Gaussian distributions

The optimal transport map between two Gaussian distributions $g_{source} = N(m_1, \Sigma_1)$ and $g_{target} = N(m_2, \Sigma_2)$ with squared Euclidean cost has a closed form expression [30] given by $T : x \mapsto m_2 + A(x - m_1)$, where $A = \Sigma_1^{-\frac{1}{2}} (\Sigma_1^{\frac{1}{2}} \Sigma_2 \Sigma_1^{\frac{1}{2}})^{\frac{1}{2}} \Sigma_1^{-\frac{1}{2}}$. We compare the proposed estimator (6) in terms of the deviation from the optimal transport map.

**Experimental setup**: We consider mean zero Gaussian distributions with unit-trace covariances. The covariance matrices are computed as $\Sigma_1 = V_1 V_1^\top / \|V_1\|_F$ and $\Sigma_2 = V_2 V_2^\top / \|V_2\|_F$, where $V_1 \in \mathbb{R}^{d \times d}$ and $V_2 \in \mathbb{R}^{d \times d}$ are generated randomly from the uniform distribution. We experiment with varying dimensions and number of data-points: $d \in \{5, 10, 50, 100, 500, 1000\}$, $m \in \{10, 20, 50, 100, 150, 200\}$, and we set $n = m$ for simplicity. For each dimension $d$, we randomly generate a source-target distribution pair. Subsequently, the source and target datasets (of size $m$) are randomly generated from their respective distributions. For a every $(d, m)$, we repeat the experiments five times and report the average mean square error (MSE) results results in Figures 1 and 2. In a second set of experiments, we also study the variance in the results of a given optimal transport problem caused due to random data-points. In this setup a source-target distribution pair is randomly generated for a given $d$. From this distribution pair, source-target datasets are randomly generated five times for every $m$. Average MSE results are reported in Tables 1-3.

**Methods**: The proposed approach employs the Gaussian kernels, $k(x, z) = \exp(-\|x - z\|^2 / 2\sigma^2)$. We chose the same $\sigma$ values for the kernels over the source and the target domains ($k_1$ and $k_2$, respectively). Initial experiments indicate that suitable values of $\sigma$ include those that does not yield high condition number of the Gram matrices (i.e, the Gram matrices are not ill-conditioned). In our setup, in general, the condition number of the Gram matrices increase with $\sigma$ for a fixed $d$ and decrease with $d$ for a fixed $\sigma$. The $\sigma$ values used in various experiments are mentioned with the results. As a baseline, we also report the results obtained from the discrete OT estimator, henceforth referred to as EMD, learned via the discrete OT problem (2).

**Evaluation**: For a given data point $x_s$ from the source distribution, a transport map estimator maps $x_s$ to a data point $x_t$ in the target distribution. Such a mapping obtained from the optimal transport

Table 1: Average MSE on the problem of learning the optimal transport map between two given multivariate Gaussian distributions with $d = 10$. For all $m$, we randomly sample data points from a fixed randomly sampled source-target distribution. We observe that the proposed approach easily outperforms EMD.

| Method | $m = 10$ | $m = 20$ | $m = 50$ | $m = 100$ | $m = 150$ | $m = 200$ |
|---|---|---|---|---|---|---|
| EMD | $0.53 \pm 0.18$ | $0.42 \pm 0.11$ | $0.32 \pm 0.06$ | $0.27 \pm 0.04$ | $0.24 \pm 0.02$ | $0.21 \pm 0.01$ |
| Proposed ($\sigma = 0.1$) | $0.45 \pm 0.14$ | $\mathbf{0.23 \pm 0.05}$ | $0.23 \pm 0.03$ | $0.19 \pm 0.03$ | $\mathbf{0.17 \pm 0.01}$ | $\mathbf{0.15 \pm 0.01}$ |
| Proposed ($\sigma = 0.2$) | $0.41 \pm 0.15$ | $0.27 \pm 0.06$ | $\mathbf{0.20 \pm 0.02}$ | $\mathbf{0.17 \pm 0.02}$ | $0.23 \pm 0.10$ | $0.16 \pm 0.04$ |
| Proposed ($\sigma = 0.3$) | $\mathbf{0.37 \pm 0.14}$ | $0.25 \pm 0.06$ | $0.22 \pm 0.03$ | $0.20 \pm 0.03$ | $0.25 \pm 0.11$ | $0.19 \pm 0.01$ |

Table 2: Average MSE on the problem of learning the optimal transport map between two given multivariate Gaussian distributions with $d = 100$. For all $m$, we randomly sample data points from a fixed randomly sampled source-target distribution. We observe that the proposed approach easily outperforms EMD.

| Method | $m = 10$ | $m = 20$ | $m = 50$ | $m = 100$ | $m = 150$ | $m = 200$ |
|---|---|---|---|---|---|---|
| EMD | $0.69 \pm 0.11$ | $0.68 \pm 0.15$ | $0.56 \pm 0.06$ | $0.45 \pm 0.03$ | $0.42 \pm 0.01$ | $0.41 \pm 0.01$ |
| Proposed ($\sigma = 0.1$) | $0.54 \pm 0.08$ | $0.54 \pm 0.10$ | $0.41 \pm 0.03$ | $0.36 \pm 0.01$ | $0.35 \pm 0.01$ | $0.34 \pm 0.08$ |
| Proposed ($\sigma = 0.3$) | $0.47 \pm 0.06$ | $0.44 \pm 0.08$ | $\mathbf{0.31 \pm 0.02}$ | $\mathbf{0.26 \pm 0.01}$ | $\mathbf{0.25 \pm 0.01}$ | $\mathbf{0.25 \pm 0.04}$ |
| Proposed ($\sigma = 0.5$) | $\mathbf{0.40 \pm 0.06}$ | $\mathbf{0.37 \pm 0.07}$ | $0.32 \pm 0.02$ | $0.29 \pm 0.01$ | $0.30 \pm 0.02$ | $0.29 \pm 0.01$ |

301 map (15) is considered as the ground truth. The proposed estimator (8) and the EMD are evaluated in
302 terms of the mean squared error (MSE) from the ground truth.

303 **Results**: The results of our first set of experiments are reported in Figures 1(a)-(f). We observe that
304 the proposed estimator obtains lower average MSE (and hence better estimation of the transport map)
305 than EMD across different number of samples $m$ and dimensions $d$. The advantage of the proposed
306 estimator over the baseline is more pronounced at higher dimension.

307 In Table 1, we report the results of the second set of experiments with $d = 10$. We again observe that
308 the proposed approach outperforms EMD. Results on the same experimental setup but with $d = 100$
309 and $d = 1000$ are report in Tables 2 and 3, respectively.

310 **Out-of-sample evaluation**: We also evaluate our estimator's ability to map out-of-sample data by
311 sampling additional $m_{oos} = 200$ points from the source distributions in the above experiments. These
312 source points are not used to learn the estimator and are only used for evaluation during the inference
313 stage. The results on out-of-sample dataset, corresponding to the first set of experiments (Figure 1)
314 are reported in Figure 2. We generate out-of-sample data points for each $(d, s)$ pair, where $d$ is the
315 dimension of the data points and $s$ is the random seed (corresponding to five repetition discussed
316 earlier). Hence, for a given $(d, s)$ pair, different estimators learned with varying $m$ are evaluated on
317 the same set of out-of-sample data points.

318 We observe that the performance on out-of-sample data points are similar to the in-sample data points
319 (Figures 1(a) & (b)). The average out-of-sample MSE generally decreases with increasing number of
320 (training) samples since a better estimator is learned with more number of (training) samples. Overall,

Table 3: Average MSE on the problem of learning the optimal transport map between two given multivariate Gaussian distributions with $d = 1000$. For all $m$, we randomly sample data points from a fixed randomly sampled source-target distribution. We observe that the proposed approach easily outperforms EMD.

| Method | $m = 10$ | $m = 20$ | $m = 50$ | $m = 100$ | $m = 150$ | $m = 200$ |
|---|---|---|---|---|---|---|
| EMD | $0.89 \pm 0.15$ | $0.64 \pm 0.11$ | $0.59 \pm 0.06$ | $0.55 \pm 0.07$ | $0.50 \pm 0.01$ | $0.49 \pm 0.01$ |
| Proposed ($\sigma = 0.1$) | $0.69 \pm 0.12$ | $0.51 \pm 0.06$ | $0.46 \pm 0.04$ | $0.43 \pm 0.03$ | $0.41 \pm 0.01$ | $0.41 \pm 0.01$ |
| Proposed ($\sigma = 0.5$) | $\mathbf{0.53 \pm 0.16}$ | $\mathbf{0.36 \pm 0.08}$ | $\mathbf{0.33 \pm 0.05}$ | $\mathbf{0.31 \pm 0.05}$ | $\mathbf{0.29 \pm 0.02}$ | $\mathbf{0.29 \pm 0.01}$ |
| Proposed ($\sigma = 1.0$) | $0.67 \pm 0.19$ | $0.54 \pm 0.12$ | $0.55 \pm 0.13$ | $0.52 \pm 0.12$ | $0.51 \pm 0.06$ | $0.50 \pm 0.03$ |

Table 4: Accuracy obtained on the target domains of the Office-Caltech dataset. The knowledge transfer to the target domain happens via in-sample source data-points, i.e., those source data-points using which the transport plan was learned.

| Task | EMD | OTLin [28] | OTKer [28] | Proposed |
|------|-----|-----------|-----------|----------|
| $A \to C$ | $80.68 \pm 1.82$ | $82.92 \pm 1.41$ | $83.07 \pm 0.63$ | $\mathbf{86.27 \pm 1.74}$ |
| $A \to D$ | $72.66 \pm 6.58$ | $82.28 \pm 5.66$ | $82.53 \pm 3.70$ | $\mathbf{84.30 \pm 5.41}$ |
| $A \to W$ | $69.05 \pm 5.08$ | $\mathbf{77.70 \pm 3.60}$ | $76.35 \pm 4.16$ | $76.22 \pm 3.32$ |
| $C \to A$ | $82.61 \pm 3.45$ | $88.31 \pm 0.94$ | $88.09 \pm 1.50$ | $\mathbf{91.05 \pm 0.79}$ |
| $C \to D$ | $68.35 \pm 10.06$ | $79.75 \pm 6.04$ | $78.99 \pm 7.95$ | $\mathbf{82.78 \pm 3.81}$ |
| $C \to W$ | $65.54 \pm 2.74$ | $71.89 \pm 3.43$ | $70.00 \pm 3.93$ | $\mathbf{74.46 \pm 4.45}$ |
| $D \to A$ | $81.50 \pm 1.99$ | $88.57 \pm 1.86$ | $85.23 \pm 1.71$ | $\mathbf{90.92 \pm 1.23}$ |
| $D \to C$ | $76.51 \pm 2.87$ | $82.17 \pm 1.70$ | $78.22 \pm 1.80$ | $\mathbf{86.84 \pm 0.86}$ |
| $D \to W$ | $91.89 \pm 2.96$ | $\mathbf{97.57 \pm 1.09}$ | $96.35 \pm 1.10$ | $96.22 \pm 1.84$ |
| $W \to A$ | $71.22 \pm 1.54$ | $80.00 \pm 1.74$ | $76.23 \pm 2.50$ | $\mathbf{86.90 \pm 2.42}$ |
| $W \to C$ | $69.55 \pm 3.18$ | $77.58 \pm 2.34$ | $73.72 \pm 2.59$ | $\mathbf{82.28 \pm 1.31}$ |
| $W \to D$ | $80.76 \pm 4.90$ | $\mathbf{97.72 \pm 1.47}$ | $96.20 \pm 2.89$ | $96.20 \pm 3.00$ |
| Average | $75.86 \pm 1.43$ | $83.87 \pm 0.38$ | $82.08 \pm 0.95$ | $\mathbf{86.20 \pm 0.98}$ |

the results illustrate the utility of the proposed approach for out-of-sample estimation. It should be noted that the baseline EMD cannot map out-of-sample data points.

## 5.2 Domain adaptation

We experiment on the Caltech-Office dataset [20], which contains images from four domains: Amazon (online retail), the Caltech image dataset, DSLR (images taken from a high resolution DSLR camera), and Webcam (images taken from a webcam). The domains vary with respect to factors such as background, lightning conditions, noise, etc. The number of examples in each domain is: 958 (Amazon), 1123 (Caltech), 157 (DSLR), and 295 (Webcam). Each domain has images from ten classes. We perform multiclass classification in the domain adaptation setting, where each domain is in turn considered as the source or the target. Overall, there are twelve adaptation tasks (e.g., task $A \to C$ has Amazon as the source and Caltech as the target domain). We employ DeCAF6 features to represent the images [11, 28, 8].

**Experimental setup**: For learning transport plan, we randomly select ten images per class for the source domain (eight per class when DSLR is the source, due to its sample size). The remaining samples of the source domain is marked as out-of-sample source data-points. The target domain is partitioned equally into training and test sets. The transport map is learned using the source-target training sets. The 'in-sample' accuracy is then evaluated on the target's test set. We also evaluate the quality of our out-of-sample estimation as follows. Instead of projecting the source training set samples onto the target domain, we project only the out-of-sample (OOS) source data-points and compute the accuracy over the target's test set. It should be noted that the transport model has not been learned on the OOS data-points, such mappings may not be as accurate as the in-sample mapping. The OOS evaluation assesses the downstream effectiveness of OOS estimation on domain adaptation. Out-of-sample estimation is especially attractive in big data and online applications. The classification in the target domain is performed using a 1-Nearest Neighbor classifier [20, 28, 8]. The above experimentation is performed five times. The average in-sample and out-of-sample accuracy are reported in Tables 4 & 5, respectively.

**Methods**: We compare our approach with EMD, OTLin [28], and OTKer [28]. Both OTLin and OTKer aim to solve the discrete optimal transport problem and also learn a transformation approximating the corresponding transport map in a joint optimization framework. OTLin learns a linear transformation while OTKer learns a non-linear transformation (e.g., via Gaussian kernel). The learned transformation allows OTLin and OTKer to perform out-of-sample estimation as well. Both OTLin and OTKer employ two regularization parameters. As suggested by their authors [28], both the regularization parameters were chosen from the set $\{10^{-3}, 10^{-2}, 10^{-1}, 10^0\}$. It should be noted that best regularization parameters were selected for each task. OTKer additionally requires Gaussian kernel's hyper-parameter $\sigma$, which was chosen from the set $\{0.1, 0.5, 1, 5, 10\}$. We use the

Table 5: Accuracy obtained on the target domains of the Office-Caltech dataset. The knowledge transfer to the target domain happens via out-of-sample source data-points, i.e., those source data-points which were not used for learning the transport plan.

| Task | OTLin [28] | OTKer [28] | Proposed |
|------|-----------|-----------|----------|
| $A \to C$ | $56.75 \pm 2.94$ | $79.11 \pm 2.76$ | $\mathbf{84.71 \pm 1.82}$ |
| $A \to D$ | $79.49 \pm 1.86$ | $82.79 \pm 2.06$ | $\mathbf{85.82 \pm 0.95}$ |
| $A \to W$ | $55.41 \pm 6.60$ | $76.35 \pm 1.66$ | $\mathbf{81.62 \pm 2.11}$ |
| $C \to A$ | $87.79 \pm 3.28$ | $84.54 \pm 2.78$ | $\mathbf{90.66 \pm 0.87}$ |
| $C \to D$ | $81.01 \pm 3.75$ | $74.94 \pm 3.53$ | $\mathbf{81.52 \pm 3.97}$ |
| $C \to W$ | $70.00 \pm 3.64$ | $68.11 \pm 1.16$ | $\mathbf{74.05 \pm 4.16}$ |
| $D \to A$ | $64.53 \pm 5.01$ | $81.95 \pm 2.69$ | $\mathbf{85.47 \pm 2.74}$ |
| $D \to C$ | $43.67 \pm 4.80$ | $72.79 \pm 3.04$ | $\mathbf{80.44 \pm 2.19}$ |
| $D \to W$ | $\mathbf{90.04 \pm 2.81}$ | $82.02 \pm 0.72$ | $88.11 \pm 2.27$ |
| $W \to A$ | $60.09 \pm 4.77$ | $73.88 \pm 2.83$ | $\mathbf{80.04 \pm 4.11}$ |
| $W \to C$ | $49.34 \pm 8.78$ | $63.17 \pm 4.14$ | $\mathbf{76.97 \pm 1.52}$ |
| $W \to D$ | $\mathbf{95.95 \pm 1.86}$ | $90.89 \pm 1.68$ | $93.42 \pm 2.45$ |
| Average | $69.51 \pm 2.70$ | $77.54 \pm 0.66$ | $\mathbf{83.60 \pm 0.22}$ |

Python Optimal Transport (POT) library (`https://github.com/PythonOT/POT`) implementations of OTLin and OTKer in our experiments. For the proposed approach, as in the previous experiments, we chose the Gaussian kernels and have same $\sigma$ values for the kernels over the source and the target domains. The $\sigma$ for our approach was also chosen from the set $\{0.1, 0.5, 1\}$.

**Results**: We observe from Tables 4 & 5 that the proposed approach outperforms the baselines, obtaining the best in-sample and out-of-sample (OOS) accuracy. As discussed, the in-sample accuracy is likely to be better than out-of-sample accuracy (for any approach). Interestingly, for a few tasks with Amazon and Caltech as the source domains, the OOS accuracy of our approach is comparable to our in-sample accuracy. In these domains, the OOS set is larger than the training set. The proposed OOS estimation is able to exploit this and provide an effective knowledge transfer. Conversely, we observe a drop in our OOS accuracy (when compared with the corresponding in-sample accuracy) in tasks with DSLR and Webcam as the source domains since the size of OOS set is quite small and hence lesser potential for knowledge transfer. On the other hand, OTLin suffers a significant drop in OOS performance, likely due the the overfitting of the learned linear transformation on the source training points. While OTKer has better OOS performance than OTLin, it has more variance between in-sample and out-of-sample performance than the proposed approach.

## 6 Conclusions

The idea of employing kernel embeddings of distributions in OT seems promising, especially in the continuous case. It not only leads to sample complexities that are dimension-free, but also provides a new regularization scheme based on MMD distances, which is complementary to existing $\phi$-divergence based regularization.

While the optimal solution of the proposed MMD regularized formulation recovers the transport plan, the objective value does not seem to have any special use. On the contrary, it has been shown that with entropic, $\phi$-divergence based regularizations the optimal objectives lead to notions of Sinkhorn divergences [13] and Hillinger-Kantorovich metrics [24]. We make an initial observation that in the special regularization, $\epsilon_i = 0, \Delta_1 = \Delta_2$, and the Tikhonov regularized form of (4), our optimal objective resembles that defining the Hillinger-Kantorovich metrics very closely. Hence, we conjecture that our optimal objective in this special case may also define a new family of metrics. However, we postpone such connections (if any) to future work.

## A Proof for Theorem 2

*Proof.* Let $\hat{h}$ denote the objective in (4), when written in Tikhonov form, as a function of variables $\mathcal{U} \in \mathcal{E}(\mathcal{H}_2, \mathcal{H}_1), \mathcal{U}_1 \in \mathcal{L}(\mathcal{H}_1, \mathcal{H}_2), \mathcal{U}_2 \in \mathcal{L}(\mathcal{H}_2, \mathcal{H}_1)$ and let $h$ denote that when the true

388  embeddings are employed instead of their estimates. In particular, we have $\hat{h}\left(\hat{\mathcal{U}},\hat{\mathcal{U}}_1,\hat{\mathcal{U}}_2\right) =$

389  $g\left(\hat{c},\hat{\mu}_s,\hat{\mu}_t,\hat{\Sigma}_{ss},\hat{\Sigma}_{tt}\right), h\left(\mathcal{U}^*,\mathcal{U}_1^*,\mathcal{U}_2^*\right) = g\left(c,\mu_s,\mu_t,\Sigma_{ss},\Sigma_{tt}\right)$, where $\hat{\mathcal{U}},\hat{\mathcal{U}}_1,\hat{\mathcal{U}}_2$ and $\mathcal{U}^*,\mathcal{U}_1^*,\mathcal{U}_2^*$

390  are optimal solutions to respective problems.

391  We begin by noting that the feasibility set of (4) is bounded. This is because: i) the set $\mathcal{E}\left(\mathcal{H}_2,\mathcal{H}_1\right)$
392  is bounded. This is true as $\mathcal{U} \in \mathcal{E}\left(\mathcal{H}_2,\mathcal{H}_1\right) \Rightarrow$ there exists $p \in \mathcal{M}^1(\mathcal{X}\times\mathcal{Y})$ such that
393  $\|\mathcal{U}\| = \|\mathbb{E}_{(X,Y)\sim p}\left[\phi_1(X)\otimes\phi_2(Y)\right]\| \le \mathbb{E}_{(X,Y)\sim p}\left[\|\phi_1(X)\otimes\phi_2(Y)\|\right] = 1$. The first inequality
394  follows from Jensens inequality and the second equality is true for any bounded kernel like Gaussian
395  and the Kroncker Delta. ii) By triangle inequality, $\left|\|\mathcal{U}\| - \|\hat{\Sigma}_{ss}\mathcal{U}_1\|\right| \le \left\|\mathcal{U} - \hat{\Sigma}_{ss}\mathcal{U}_1^\top\right\| \le \epsilon_1$.
396  This shows that the set of all feasible $\hat{\Sigma}_{ss}\mathcal{U}_1$ is bounded, since $\mathcal{U}$ is itself bounded in the
397  feasibility set. Now, since $maxeig(\hat{\Sigma}_{ss}) = maxeig(G_2)/n \le tr(G_2)/n = 1$ (again
398  true for Kronecker and Gaussian kernels), we obtain that set of all feasible $\mathcal{U}_1$ is also
399  bounded. Similarly, set of all feasible $\mathcal{U}_2$ is bounded. Accordingly, we define $\mathcal{B}\left(\epsilon_1,\epsilon_2\right) \equiv$
400  $\{(\mathcal{U} \in \mathcal{E}\left(\mathcal{H}_2,\mathcal{H}_1\right),\mathcal{U}_1 \in \mathcal{L}\left(\mathcal{H}_1,\mathcal{H}_2\right),\mathcal{U}_2 \in \mathcal{L}\left(\mathcal{H}_2,\mathcal{H}_1\right)) \mid \|\mathcal{U}\| \le 1, \|\mathcal{U}_1\| \le 1+\epsilon_1, \|\mathcal{U}_2\| \le 1+\epsilon_2\}$.
401  By the above argument, it is clear that there is no loss of generality in further restrict-
402  ing the search space to that with intersection with this bounded set, $\mathcal{B}\left(\epsilon_1,\epsilon_2\right)$, and always
403  $(\mathcal{U}^*,\mathcal{U}_1^*,\mathcal{U}_2^*),\left(\hat{\mathcal{U}},\hat{\mathcal{U}}_1,\hat{\mathcal{U}}_2\right) \in \mathcal{B}\left(\epsilon_1,\epsilon_2\right)$ for any $m,n \in \mathbb{N}$.

404  The rest of the proof follows from the claim below:

**Claim 1.** *The uniform bound:*

$$\max_{(\mathcal{U},\mathcal{U}_1,\mathcal{U}_2)\in\mathcal{B}(\epsilon_1,\epsilon_2)} \left|\hat{h}\left(\mathcal{U},\mathcal{U}_1,\mathcal{U}_2\right) - h\left(\mathcal{U},\mathcal{U}_1,\mathcal{U}_2\right)\right| \le O\left(1/\sqrt{\min(m,n)}\right)$$

405  *holds, where the constants in the RHS are dimension-free.*

406  *Proof.* Now consider the Tikhonov regularized form of (4). Then, one of the term in the objective is
407  $\|\mathcal{U}_1\hat{\mu}_s - \hat{\mu}_t\| \le \|\mathcal{U}_1(\hat{\mu}_s - \mu_s)\| + \|\mu_t - \hat{\mu}_t\| + \|\mathcal{U}_1\mu_s - \mu_t\|$, which is less than $\|\mathcal{U}_1\mu_s - \mu_t\| + O(\frac{1}{\sqrt{p}})$
408  with high probability. Here, $p = \min(m,n)$. The first inequality is by triangle inequality and the
409  second is the crucial one that follows from sample complexity of kernel mean embeddings (see
410  theorem 2 in [36]), and the boundedness of $\|\mathcal{U}_1\|$. Also, the constants in $O(\frac{1}{\sqrt{p}})$ are independent
411  of samples, variables and dimensions. By symmetry, we also have with high probability that
412  $\|\mathcal{U}_1\mu_s - \mu_t\| \le \|\mathcal{U}_1\hat{\mu}_s - \hat{\mu}_t\| + O(\frac{1}{\sqrt{p}})$. Hence, with high probability, uniformly over the feasibility
413  set, $\|\|\mathcal{U}_1\mu_s - \mu_t\| - \|\mathcal{U}_1\hat{\mu}_s - \hat{\mu}_t\|\| \le O(\frac{1}{\sqrt{p}})$. Analogous arguments hold for the other quadratic
414  terms too. Now, we analyze the linear objective term. By Jensen's inequality, $|\langle\hat{c},\mathcal{U}\rangle - \langle c,\mathcal{U}\rangle| \le$
415  $\sqrt{\mathbb{E}_u\left[(\hat{c}-c)^2\right]}$, where $u$ is the measure corresponding to $\mathcal{U}$. Let $\bar{u}$ denote the product measure
416  of the given marginals. It is easy to see that $\{(x_i,y_j) \mid i \in 1,\ldots,m, j \in 1,\ldots,n\}$ is an iid set of
417  samples from $\bar{u}$. By eqn. (4) in theorem 3.1 in [31] and lemma 1 in [32], we have $\sqrt{\mathbb{E}_u\left[(\hat{c}-c)^2\right]} \le$
418  $\frac{\|c\|_{\bar{u}}}{\sqrt{mn}}\left(1 + \sqrt{2\log(\frac{1}{\delta})}\right)$, with probability $\delta$. Here, $\|\cdot\|_{\bar{u}}$ is same as that defined in section III of [31],
419  and theorem 3.1 in [31] applies to our case as we assumed normalized kernels. In particular, this
420  bound is independent of dimensions and $\mathcal{U}$. To summarize, we have, $|\langle\hat{c},\mathcal{U}\rangle - \langle c,\mathcal{U}\rangle| \le O(\frac{1}{\sqrt{mn}})$.
421  Finally, again by triangle inequality, $\left|\hat{h}\left(\mathcal{U},\mathcal{U}_1,\mathcal{U}_2\right) - h\left(\mathcal{U},\mathcal{U}_1,\mathcal{U}_2\right)\right|$ is less than the sum of deviations
422  in each of the terms detailed above. Since each of these deviations is upper bounded uniformly by
423  $O\left(\frac{1}{\sqrt{p}}\right)$, the claim is proved. $\square$

424  The proof of the theorem then follows from standard arguments: $\hat{h}\left(\hat{\mathcal{U}},\hat{\mathcal{U}}_1,\hat{\mathcal{U}}_2\right) -$
425  $h\left(\mathcal{U}^*,\mathcal{U}_1^*,\mathcal{U}_2^*\right) \le h\left(\hat{\mathcal{U}},\hat{\mathcal{U}}_1,\hat{\mathcal{U}}_2\right) - h\left(\mathcal{U}^*,\mathcal{U}_1^*,\mathcal{U}_2^*\right) + \max_{(\mathcal{U},\mathcal{U}_1,\mathcal{U}_2)\in\mathcal{B}(\epsilon_1,\epsilon_2)}\hat{h}\left(\mathcal{U},\mathcal{U}_1,\mathcal{U}_2\right) -$
426  $h\left(\mathcal{U},\mathcal{U}_1,\mathcal{U}_2\right) \le h\left(\hat{\mathcal{U}},\hat{\mathcal{U}}_1,\hat{\mathcal{U}}_2\right) - h\left(\mathcal{U}^*,\mathcal{U}_1^*,\mathcal{U}_2^*\right) + O\left(1/\sqrt{\min(m,n)}\right)$ by the claim.
427  Now, the estimation error, $h\left(\hat{\mathcal{U}},\hat{\mathcal{U}}_1,\hat{\mathcal{U}}_2\right) - h\left(\mathcal{U}^*,\mathcal{U}_1^*,\mathcal{U}_2^*\right)$, which is non-negative,

428 is equal to $\left(\hat{h}\left(\hat{\mathcal{U}},\hat{\mathcal{U}}_1,\hat{\mathcal{U}}_2\right) - \hat{h}\left(\mathcal{U}^*,\mathcal{U}_1^*,\mathcal{U}_2^*\right)\right)$ + $\left(h\left(\hat{\mathcal{U}},\hat{\mathcal{U}}_1,\hat{\mathcal{U}}_2\right) - \hat{h}\left(\hat{\mathcal{U}},\hat{\mathcal{U}}_1,\hat{\mathcal{U}}_2\right)\right)$ +

429 $\left(\hat{h}\left(\mathcal{U}^*,\mathcal{U}_1^*,\mathcal{U}_2^*\right) - h\left(\mathcal{U}^*,\mathcal{U}_1^*,\mathcal{U}_2^*\right)\right)$ $\leq$ $O\left(1/\sqrt{\min(m,n)}\right)$. The last inequality follows

430 from the claim and the definition of $\left(\hat{\mathcal{U}},\hat{\mathcal{U}}_1,\hat{\mathcal{U}}_2\right)$ that it minimizes $\hat{h}$. Analogous arguments give

431 $h\left(\mathcal{U}^*,\mathcal{U}_1^*,\mathcal{U}_2^*\right) - \hat{h}\left(\hat{\mathcal{U}},\hat{\mathcal{U}}_1,\hat{\mathcal{U}}_2\right) \leq O\left(1/\sqrt{\min(m,n)}\right)$. This not only completes the proof but also

432 shows that the estimation error also decays with rate that is dimension-free. $\qquad\square$

## B  Proof of representer theorem

434 *Proof.* Without loss of generality, we consider the parameterization: $\mathcal{U}^\alpha = \sum_{i=1}^m \sum_{j=1}^n \alpha_{ij}\phi_1(x_i)\otimes$

435 $\phi_2(y_j)+\mathcal{U}^\perp, \mathcal{U}_1^\beta = \sum_{i=1}^m \sum_{j=1}^n \beta_{ji}\phi_2(y_j)\otimes\phi_1(x_i)+\mathcal{U}_1^\perp, \mathcal{U}_2^\gamma = \sum_{i=1}^m \sum_{j=1}^n \gamma_{ij}\phi_1(x_i)\otimes\phi_2(y_j)+$

436 $\mathcal{U}_2^\perp$, where $\mathcal{U}^\perp, \mathcal{U}_1^\perp, \mathcal{U}_2^\perp$ are the respective orthogonal complements. It is easy to see that the objective

437 as well as the first two inequalities in (4) do not involve the orthogonal complements. Also the term

438 $\left\|\mathcal{U} - \hat{\Sigma}_{ss}\mathcal{U}_1^\top\right\|^2_{\mathcal{H}_2\otimes\mathcal{H}_1}$ can be written as sum of a term not involving the orthogonal complements

439 and $\|\mathcal{U}^\perp - \hat{\Sigma}_{ss}\left(\mathcal{U}_1^\perp\right)^\top\|^2_{\mathcal{H}_2\otimes\mathcal{H}_1}$. Like-wise $\left\|\mathcal{U} - \mathcal{U}_2\hat{\Sigma}_{tt}\right\|^2_{\mathcal{H}_2\otimes\mathcal{H}_1}$ can be written as sum of a term

440 not involving the orthogonal complements as $\|\mathcal{U}^\perp - \mathcal{U}_2^\perp\hat{\Sigma}_{tt}\|^2_{\mathcal{H}_2\otimes\mathcal{H}_1}$.

441 Now re-writing (4), where all the norm constraints are equivalently replaced by the norm-

442 squared constraints, in Tikhonov regularization form reads as: $\min_{f\in\mathcal{S}\subseteq\mathcal{H}} \hat{\mathcal{R}}[f] + \Omega[f]$, where

443 $f = (\mathcal{U},\mathcal{U}_1,\mathcal{U}_2), \mathcal{H} = (\mathcal{H}_2\otimes\mathcal{H}_1)\oplus(\mathcal{H}_1\otimes\mathcal{H}_2)\oplus(\mathcal{H}_2\otimes\mathcal{H}_1), \mathcal{S} = \mathcal{E}(\mathcal{H}_2,\mathcal{H}_1)\times\mathcal{L}(\mathcal{H}_1,\mathcal{H}_2)\times$

444 $\mathcal{L}(\mathcal{H}_2,\mathcal{H}_1), \Omega[f] \equiv \|\mathcal{U}^\perp - \hat{\Sigma}_{ss}\left(\mathcal{U}_1^\perp\right)^\top\|^2_{\mathcal{H}_2\otimes\mathcal{H}_1} + \|\mathcal{U}^\perp - \mathcal{U}_2^\perp\hat{\Sigma}_{tt}\|^2_{\mathcal{H}_2\otimes\mathcal{H}_1}$ and $\hat{\mathcal{R}}[f]$ is the re-

445 maining objective that does not involve the orthogonal complements. Also, let $\hat{\mathcal{S}} \subset \mathcal{S}$ denote

446 $\left\{f = (\mathcal{U},\mathcal{U}_1,\mathcal{U}_2) \in \mathcal{S} \mid \mathcal{U}^\perp = 0, \mathcal{U}_1^\perp = 0, \mathcal{U}_2^\perp = 0\right\}$ and let $\Pi_{\hat{\mathcal{S}}}$ denote the projection onto $\hat{\mathcal{S}}$. Now,

447 for any $f \in \mathcal{H}$, we have that: $\hat{\mathcal{R}}[\Pi_{\hat{\mathcal{S}}}(f)] = \hat{\mathcal{R}}[f]$ and more importantly, $0 = \Omega[\Pi_{\hat{\mathcal{S}}}(f)] \leq \Omega[f]$.

448 Consider the following argument[1]: $\min_{f\in\mathcal{S}\subset\mathcal{H}} \hat{\mathcal{R}}[f] + \Omega[f] \leq \min_{f\in\hat{\mathcal{S}}\subset\mathcal{H}} \hat{\mathcal{R}}[f] + \Omega[f] =$

449 $\min_{f\in\mathcal{S}\subset\mathcal{H}} \hat{\mathcal{R}}[\Pi_{\hat{\mathcal{S}}}(f)] + \Omega[\Pi_{\hat{\mathcal{S}}}(f)] \leq \min_{f\in\mathcal{S}\subset\mathcal{H}} \hat{\mathcal{R}}[f] + \Omega[f]$. This proves that the orthogonal

450 complements are all zero at optimality.

451 Now, let $\mathcal{L} \equiv \{\mathcal{U}^\alpha \mid \alpha \in \mathbb{R}^{m\times n}\}$, $\mathcal{P} \equiv \{\mathcal{U}^\alpha \mid \alpha \in \mathbb{R}^{m\times n}, \alpha \geq 0, \mathbf{1}^\top\alpha = 1\}$ and $\mathcal{A} \equiv$

452 $\left\{\sum_{i=1}^{m'}\sum_{j=1}^{n'} \alpha_{ij}\phi_1(x_i')\otimes\phi_2(y_j') \mid \alpha \in \mathbb{R}^{m'\times n'}, \alpha \geq 0, \mathbf{1}^\top\alpha = 1, x_i' \in \mathcal{X}, y_j' \in \mathcal{Y}, m',n' \in \mathbb{N}\right\}$.

453 Then, the only thing left to be shown is that $\mathcal{E}(\mathcal{H}_2,\mathcal{H}_1)\cap\mathcal{L} = \mathcal{P}$. While $\mathcal{P} \subseteq \mathcal{E}(\mathcal{H}_2,\mathcal{H}_1)\cap\mathcal{L}$ is

454 trivial. The converse is true because of the following facts:

455     1. $\mathcal{E}(\mathcal{H}_2,\mathcal{H}_1) = cl(\mathcal{A})$, where $cl$ denotes the set closure. While $cl(\mathcal{A}) \subseteq \mathcal{E}(\mathcal{H}_2,\mathcal{H}_1)$ is trivial,
456     the converse follows from the convergence of average of sample embeddings to the true
457     embedding (see theorem 2 in [36]).

458     2. if $\mathcal{U} \in \mathcal{L}\backslash\mathcal{P}$, then $\mathcal{U} \notin \mathcal{A}$. This is because the expansion of embeddings in RKHS of a
459     universal kernel are unique. Also, $\mathcal{U} \in \mathcal{L}, \mathcal{U} \notin \mathcal{A} \Rightarrow \mathcal{U} \notin cl(\mathcal{A})$.

460     $\qquad\square$

## C  Proof of Theorem 4

462 *Proof.* Firstly, by theorem 2, for low enough hyper-parameters we know that the conditional operator

463 obtained by solving (5) are consistent with dimension-free sample complexity. Hence the Barycentric-

464 projection problem is nothing but a stochastic optimization problem with samples as $y_j$ with likelihood

465 $\sum_{j=1}^n \left(\beta_{ji}^* k_1(x_i,x)\right)$. Using sampling with replacement, these can be converted to $m'$ iid samples

466 with uniform likelihood. Since the cost is assumed to be a metric or it's power greater than unity,

467 the stochastic optimization problem is infact convex wrt $y$. Since the domain $\mathcal{Y}$ is bounded, it is

also Lipschitz continuous wrt. $y$. Hence by (7), theorem 3 in [35], the estimation error in optimal transport map when solved by SGD is $O(1/\sqrt{m'})$ and remains dimension-free. $\qquad\square$

## Footnotes

[1]See also [3] for similar a argument.

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

[Supplementary Material 2]

# USING SEDUMI 1.02, A MATLAB*TOOLBOX FOR OPTIMIZATION OVER SYMMETRIC CONES†
## (Updated for Version 1.05)

JOS F. STURM‡

Department of Econometrics, Tilburg University,
Tilburg, The Netherlands.
http://fewcal.kub.nl/~sturm, E-mail: j.f.sturm@kub.nl.

August 1998 – October 2001

**Abstract**

SeDuMi is an add-on for MATLAB, which lets you solve optimization problems with linear, quadratic and semidefiniteness constraints. It is possible to have complex valued data and variables in SeDuMi. Moreover, large scale optimization problems are solved efficiently, by exploiting sparsity. This paper describes how to work with this toolbox.

Keywords: Symmetric cone, semidefinite programming, second order cone programming, self–duality, MATLAB, SeDuMi

SeDuMi stands for Self-Dual-Minimization: it implements the *self-dual* embedding technique for optimization over *self-dual* homogeneous cones. The self-dual embedding technique as proposed by Ye, Todd and Mizuno [31], essentially makes it possible to solve certain optimization problems in a single phase, leading either to an optimal solution, or a certificate of infeasibility. Optimization over self–dual homogeneous cones, or more concisely, optimization over symmetric cones, was first studied by Nesterov and Todd [21], and is currently an active area of research.

Semidefinite programming is a special case of optimization over symmetric cones. The popular package SP by Vandenberghe and Boyd [29] is one of the first software tools that was developed for semidefinite programming. Some control theorists use SP indirectly via LMI-TOOL, by El Ghaoui, Nikoukhah and Delebecque [8], or MRCT, by Dussy and El Ghaoui [6],

which are user-friendly front-ends for SP. More recently, the user-friendly front-end *SeDuMi Interface*[22] is available from

http://www.laas.fr/~peaucell/SeDuMiInt.html

A more recent and faster solver for semidefinite programming is SDPA, by Fujisawa, Kojima and Nakata [11]. Other solvers for semidefinite programming are CSDP by Borchers [2], SDPHA by Brixius and Potra [3] and SDPT3 by Toh, Todd and Tütüncü [28]. See Mittelmann [19] for a comparison of the performance of various solvers (including SeDuMi) on semidefinite programming problems of the SDPLIB test set.

For optimization over symmetric cones, there were until recently only two software tools available, viz. SDPPack, by Alizadeh et al. [1], and SeDuMi. (SDPT3 has been extended to this class of problems in 2001.) Both operate under the MATLAB environment, so that they can easily be used within specific applications. SeDuMi has some features that are not available in SDPPack, namely it

- allows the use of complex valued data,

- generates Farkas-dual solutions for infeasible problems,

- takes full advantage of sparsity, leading to significant speed benefits,

- has a theoretically proven $O(\sqrt{n}\log(1/\epsilon))$ worst-case iteration bound,

- promotes sparsity by handling dense columns separately (since version 1.03), using a technique proposed by Goldfarb and Scheinberg [13],

- can import linear programs in MPS format (either using a link to LIPSOL [32] or by using the loadmps add-on by the author), and semidefinite programs in SDPA [11] format.

It is also possible to convert optimization problems from SDPPack [1] format into SeDuMi. Notice that earlier versions of the semidefinite programming solver SDPT3 were also able to handle complex valued data [28], but this option has been removed recently. The issue of expoiting sparsity in semidefinite programming was studied by Fujisawa, Kojima and Nakata [10]. Unlike the approach of [10], SeDuMi uses always the same sparsity expoiting procedure to form the normal equations; this procedure is efficient regardless of the degree of sparsity. See Ross [24] for a comparison between SDPpack and SeDuMi.

The remainder of this document is a step-by-step tutorial for SeDuMi. The on-line help pages serve as a reference to the toolbox. In addition, the Appendix to this document has the details of the calling sequence for the main function, sedumi.

# 1 introduction to sedumi

Throughout this document, we assume that SeDuMi is correctly installed, and that you are working under MATLAB Version 5 or later. Entering the MATLAB command 'help' should produce a list of all installed MATLAB Toolboxes, including the following lines:

```
>> help

HELP topics:

matlab/SeDuMi105        - SeDuMi 1.05    (OCT2001)
SeDuMi105/conversion    - Conversion to SeDuMi.
```

For more help on directory/topic, type "help topic".

The command `help conversion` produces a list of functions for importing data into SeDuMi. This includes an 'umbrella' script, **getproblem**[1], which works as follows:

```
>> pname = 'truss2'; getproblem, who

Your variables are:

At        MATNAME   b         pname
K         PROBDIR   c
```

This imports problem '`truss2`', and places it in the variables `At`, `b`, `c` and `K`. To do this, SeDuMi must be able to find the requested problem somewhere on your disk. It can locate sparse SDPA problems, if you have assigned a UNIX or DOS environment variable 'SDPLIB' to the directory path where SDPA problems are stored. (SDPLIB and `SeDuMi` have a different canonical form; if $y$ is a dual optimal solution calculated by `SeDuMi`, then $-y$ is an optimal solution in the SDPLIB canonical form.) If LIPSOL is installed, it uses LIPSOL's function `findprob` to locate linear programming problems in MPS format. Finally, if SDPPack is installed, and the environment variable `SDPPACK` points to the SDPPack directory, then SDPPack problems are searched for in the directory 'SDPPACK/problems'.

Typing '`help SeDuMi105`' produces a list of the functions that you can use to build and solve optimization models over symmetric cones. They are: `sedumi`, `eigK`, `vec`, `mat` and `eyeK`. Online help is provided by `help sedumi`, `help eigK`, and so on. The following sections give a more detailed explanation of these functions, with some illustrating examples.

## 2  linear programming

It is possible to formulate your linear programming model in either the primal standard form,

$$
\begin{aligned}
\text{minimize} \quad & c^{\mathrm{T}}x \\
\text{such that} \quad & Ax = b \\
& x_i \geq 0 \text{ for } i = 1, 2, \ldots, n,
\end{aligned}
\tag{1}
$$

or the dual standard form,

$$
\begin{aligned}
\text{maximize} \quad & b^{\mathrm{T}}y \\
\text{such that} \quad & c_i - a_i^{\mathrm{T}}y \geq 0 \text{ for } i = 1, 2, \ldots, n.
\end{aligned}
\tag{2}
$$

Mixed form and symmetric form linear programming models may be formulated using the 'free variable' definition in SeDuMi, as outlined in Section 2.1.

Suppose that you want to solve the following linear programming problem:

$$
\begin{aligned}
\text{minimize} \quad & x_1 - x_2 \\
\text{such that} \quad & 10x_1 - 7x_2 \geq 5 \\
& x_1 + x_2/2 \leq 3 \\
& x_1 \geq 0, \ x_2 \geq 0.
\end{aligned}
\tag{3}
$$

In order to formulate this LP problem in the primal standard form, we have to add slack variables, say $x_3$ and $x_4$. In MATLAB, we can then enter the $b$ and $c$ vectors, and the $A$ matrix as follows:

```
>>  c = [1; -1; 0; 0];
>>  A = [10, -7, -1, 0; 1, 1/2, 0, 1];
>>  b = [5; 3];
```

We can now solve problem (3) in the primal form (1) by invoking the function sedumi. Remark that MATLAB is case sensitive, and it is therefore essential to write sedumi in lower case.

```
>> sedumi(A,b,c)
SeDuMi by Jos F. Sturm, 1998.
Alg = 2: xz-corrector, theta = 0.250, beta = 0.500
eqs m = 2, order n = 5, dim = 5, blocks = 1
 it :      cx         gap    delta  rate  t/maxt    feas
  0 :            5.00E+00 0.000
  1 :   7.81E-01 9.79E-01 0.000 0.1959 0.9000    0.77
  2 :  -5.52E-02 9.40E-02 0.000 0.0959 0.9900    0.93
```

```
* 3 : -1.25E-01 5.70E-04 0.000 0.0061 0.9990   1.08
iter seconds digits       c*x             b*y
  3      0.1  15.1 -1.2500000000e-01 -1.2500000000e-01
|Ax-b| =   1.8e-15, |x|=  2.9e+00, |y|=  2.8e-01

ans =

    1.9583
    2.0833
         0
         0
```

This shows that the optimal value is $-0.125$, as listed under `c*x`. The function `sedumi` returns an optimal solution, which in this case is $x_1 = 1.9583$ and $x_2 = 2.0833$. Notice that $x$ is indeed feasible for (1), because all its components are nonnegative, and $Ax = b$, as can be checked by the commands `min(x)` and `norm(Ax-b)`, respectively. Of course, some round-off errors may occur, as can be seen from the following MATLAB output:

```
>> A=sparse(A);norm(A*x-b)

ans =

    1.7764e-15

>> norm(A*(24*x)-24*b)

ans =

     0
```

The first quantity is listed as $|Ax - b| = 1.8e - 15$ in the output of SeDuMi. The second line shows that the reported value of `|Ax-b|` does not only contain the residual, but also errors in computing the residual. The meaning of the other parts in the output of SeDuMi is explained in Section 5.

Using dual solutions, it is possible to check also optimality. Namely, if we let $z := c - A^{\mathrm{T}}y$, then if $x$ and $y$ are feasible to (1) and (2) respectively, we have

$$0 \leq z^{\mathrm{T}}x = c^{\mathrm{T}}x - y^{\mathrm{T}}Ax = c^{\mathrm{T}}x - b^{\mathrm{T}}y.$$

Thus, if $c^{\mathrm{T}}x - b^{\mathrm{T}}y = 0$, then $c^{\mathrm{T}}x$ must be minimal, and $b^{\mathrm{T}}y$ must be maximal, over all feasible solutions. The dual solution $y$ to (2) is assigned to the second output argument of `sedumi`, as in

```
>> [x,y] = sedumi(A,b,c)
```

In this example, we have $y_1 = 0.125$ and $y_2 = 0.25$. Issueing the command

```
>> z = c - A'*y
z =

        0
        0
   0.1250
   0.2500
```

we see that $z_i x_i = 0$ for all $i$, proving optimality. However, due to some round-off errors, $c^{\mathrm{T}}x - b^{\mathrm{T}}y$ is positive in this case. The quantity `digits = 15.1` in the output of `SeDuMi`, is defined as follows:

$$
\mathtt{digits} = \begin{cases} -\log_{10}((c^{\mathrm{T}}x - b^{\mathrm{T}}y)/(|b^{\mathrm{T}}y| + 10^{-10})) \text{ if } c^{\mathrm{T}}x - b^{\mathrm{T}}y > 0 \\ \infty \text{ otherwise.} \end{cases} \tag{4}
$$

As is well known, $y$ is a subgradient of the optimal value function in terms of changes in $b$. If the optimal value function is locally not differentiable in $b$, i.e. if there are multiple dual optimal solutions, then it is said to be primal degenerate. `SeDuMi` usually generates a solution $y$ in the relative interior of the subgradient set, because it uses a Mehrotra-Ye [18] type termination procedure for linear programs. For a detailed treatment of sensitivity analysis based on such solutions, we refer to Monteiro and Mehrotra [20] and the book of Roos, Terlaky and Vial [23].

For large problems, it is usually not feasible to store $A$ as a full matrix, due to memory limitations. In this case, $A$ should be stored in sparse format; type `help sparfun` for details. Internally, `SeDuMi` always converts $A$ to sparse format. The $b$ and $c$ vectors can also be in sparse format, if desired.

In the preceding, we defined $b$ and $c$ in MATLAB as column vectors, but this is not essential; `SeDuMi` produces the same output if $b$ and/or $c$ are defined as row vectors. Similarly, `SeDuMi` is not picky about the orientation of $A$: it will detect the correct orientation based on the $b$ and $c$ vectors (except in the unrealistic case that $A$ is square). In fact, it is good practice to store $A$ in such a way that it has more rows than columns, which is the transpose orientation of the $A$ matrices that we have seen so far. Namely, if $A$ is stored in sparse format, then it is stored as a set of sparse column vectors. Hence, if there are fewer columns, it will occupy less space.

There is a third output argument of `SeDuMi`, called `info`. In our example,

```
>> [x,y,info]=sedumi(A,b,c); info

info =
```

```
    cpusec: 0.1100
      iter: 3
 feasratio: 1
    numerr: 0
      pinf: 0
      dinf: 0
```

This is a compound output argument, or structure, with a field `cpusec` for the solution time, `iter` for the number of iterations, a field `numerr` which is nonzero in case of numerical problems (1 means premature termination: results are inaccurate, 2 means failure), a field `feasratio` for the final value of the feasibility indicator, and two fields, `pinf` and `dinf`, for the detected feasibility status of the optimization problem. If `pinf` = 1, then the primal problem (1) is infeasible, and $y$ is a Farkas dual solution.

For instance, if we change the $b$ vector in the preceding example to $b = \begin{bmatrix} 5, & 0.4 \end{bmatrix}$, then SeDuMi yields `info.pinf` = 1, $b^{\mathrm{T}}y = 0.0955 > 0$, $\max_{i=1,2,3,4} a_i^{\mathrm{T}} y = -0.1866 \leq 0$. Notice that for any $x$ with $Ax = b$, we have $y^{\mathrm{T}}Ax = b^{\mathrm{T}}y = 0.0955 > 0$, whereas $y^{\mathrm{T}}Ax \leq 0$ for nonnegative $x$, because all components of $A^{\mathrm{T}}y$ are nonpositive. A Farkas dual solution thus provides a certificate of infeasibility. In this example, there appears to be a Farkas dual solution for which all entries of $A^{\mathrm{T}}y$ are strictly negative. In general though, they are merely nonpositive. For numerical reasons, $A^{\mathrm{T}}y$ can then contain some small positive components, and in this case we have an approximate Farkas dual solution. Loosely speaking, such solutions demonstrate that there cannot be any reasonably sized primal feasible solution; see Todd and Ye [27] for details.

Suppose now that we want to solve a problem in the dual standard form (2). In this case, $y$ with $b^{\mathrm{T}}y > 0$ and $A^{\mathrm{T}}y \leq 0$ has the interpretation of an improving direction. Namely, if there exists a feasible solution $\bar{y}$, i.e. if $c - A^{\mathrm{T}}y \geq 0$, then $\bar{y} + ty$ is feasible for all $t \geq 0$, and $\lim_{t \to \infty} b^{\mathrm{T}}y = \infty$. In this case, we say that the problem is unbounded. The other possibility is that there does not exist any feasible solution $\bar{y}$, i.e. the problem is infeasible. To distinguish between an infeasible and an unbounded problem, we have to go through a second stage, by solving a feasibility problem:

```
>> [x,y,info] = sedumi(A,zeros(length(b),1),c)
```

which may be entered in simplified form (since version 1.05) as

```
>> [x,y,info] = sedumi(A,0,c)
```

In our previous example, the dual problem is feasible, and the above command yields a feasible solution $\bar{y}$. The need for this second stage feasibility problem is typical for interior point methods with the self–dual embedding technique of Ye, Todd and Mizuno [31].

The interpretation of `info.dinf` is analogous to that of `info.pinf`. Namely, if `info.dinf = 1` then the dual problem (2) is infeasible, and this claim is certified by a Farkas solution $x$ with

$$c^{\mathrm{T}}x < 0,\ Ax = 0,\ x_i \geq 0 \text{ for } i = 1, 2, \ldots, n.$$

To distinguish between primal unboundedness and primal infeasibility, we would then solve the feasibility problem

```
>> [x,y,info] = sedumi(A,b,zeros(length(c),1))
```

SeDuMi can also generate Gordan-Stiemke dual solutions. For instance, if we restore the vector $b$ to $b^{\mathrm{T}} = \begin{bmatrix} 5, & 3 \end{bmatrix}$, and solve the feasibility problem `x = sedumi(A,b,zeros(4,1))`, we obtain a strictly positive vector $x$. This is because interior point methods try to find a solution in the relative interior of the solution set. To see what happens if feasible solutions can merely be nonnegative, consider the following example:

```
>> b = [5, 1/2];
>> [x,y,info]=sedumi(A,b,zeros(4,1));
>> [x -A'*y]

ans =

    0.5000         0
         0    1.1875
         0    0.0990
         0    0.9895

>> b*y

ans =

    1.3878e-17
```

In this example, the primal does not have an interior solution, i.e. it is weakly feasible, and this is demonstrated by a Gordan-Stiemke dual solution $y$. Namely, $0 \neq A^{\mathrm{T}}y \leq 0$, and $b^{\mathrm{T}}y = 0$, which clearly implies that there cannot be any $x > 0$ such that $A * x = b$.

SeDuMi treats the primal and dual in a symmetric way, i.e. it does not favor one over the other. From a modeling point of view however, the primal standard form and the dual standard form are quite different, and it depends on the application which one is more favorable. The primal form has the advantage of explicit equality constraints. In principle, equality constraints can be constructed in the dual form also, simply by means of two inequality constraints, such as

$a_i^{\mathrm{T}} y \leq c_i$ and $a_i^{\mathrm{T}} y \geq c_i$. However, this technique is not recommended, since such constraint pairs tend to get a pair of very large primal multipliers $x_i$, hence leading to numerical difficulties. It may be better to enforce an equality constraint by eliminating a $y$ variable. However, the latter technique may destroy the sparsity structure of the $A$-matrix, thus leading to longer solution times.

Exactly the same problems arise in modeling a free (i.e. unresistricted in sign) variable in the primal standard form. Splitting such a variable into two, its positive part and its negative part, often results in numerical difficulties. One may also try to eliminate such a variable by removing an equality constraint, but this usually causes an increase in the number of nonzeros in the $A$-matrix. An alternative is to model all free variables in a quadratic cone. Quadratic cones are discussed in Section 3. To prevent numerical difficulties with this technique, it is desirable to fix a – possibly large – upper bound on the norm of the vector of free variables, which is easily done in a quadratic cone.

Since Version 1.05, the user does not need to worry about these issues, since free variables are allowed. This is the topic of the section below.

## 2.1 Free variables

It is possible to formulate your linear programming model in a primal form with free variables as follows:

$$
\begin{aligned}
&\text{minimize} && c^{\mathrm{T}} x \\
&\text{such that} && Ax = b \\
&&& x_i \in \Re \text{ for } i = 1, 2, \ldots, \texttt{K.f}, \\
&&& x_j \geq 0 \text{ for } j = \texttt{K.f} + 1, \texttt{K.f} + 2, \ldots, n,
\end{aligned}
\tag{5}
$$

where $\texttt{K.f}$ is the number of *free variables*. The associated slack variables in the dual problem are then restricted to be zero, thus allowing *equality constraints* in the dual:

$$
\begin{aligned}
&\text{maximize} && b^{\mathrm{T}} y \\
&\text{such that} && c_i - a_i^{\mathrm{T}} y = 0 \text{ for } i = 1, 2, \ldots, \texttt{K.f} \\
&&& c_j - a_j^{\mathrm{T}} y \geq 0 \text{ for } j = \texttt{K.f} + 1, \texttt{K.f} + 2, \ldots, n
\end{aligned}
\tag{6}
$$

# 3 quadratic and semidefinite constraints

In SeDuMi, it is possible to impose quadratic or semidefinite constraints, by restricting variables to a quadratic cone or the cone of positive semidefinite matrices, respectively. Such a restriction then replaces the nonnegativity restriction in linear programming. Thus, instead of requiring $x \in \Re_+^n$ as in (1), we will now require $x \in \mathcal{K}$, where $\mathcal{K}$ is a so-called symmetric cone. A symmetric cone is a Cartesian product of a nonnegative orthant, quadratic cones and cones of

positive semidefinite matrices. The standard primal form for such optimization problems is

$$\begin{aligned} \text{minimize} \quad & c^{\mathrm{T}}x \\ \text{such that} \quad & Ax = b \\ & x \in \mathcal{K} \end{aligned} \tag{7}$$

and the dual standard form is

$$\begin{aligned} \text{maximize} \quad & b^{\mathrm{T}}y \\ \text{such that} \quad & c - A^{\mathrm{T}}y \in \mathcal{K}. \end{aligned} \tag{8}$$

## 3.1    The quadratic cone

A quadratic cone is by definition a cone of the form

$$\text{Qcone} := \{(x_1, x_2) \in \Re \times \Re^{N-1} \mid x_1 \geq \|x_2\|\}, \tag{9}$$

where $\|\cdot\|$ denotes the Euclidean norm (the function `norm` in MATLAB). The quadratic cone is also known as the second order cone or Lorentz cone. As an example, consider the following optimization problem:

$$\min\left\{y_1 + y_2 \,\middle|\, y_1 \geq \|q - Py_3\|,\, y_2 \geq \sqrt{1 + \|y_3\|^2}\right\}, \tag{10}$$

where $P$ is a given matrix, and $q$ a given vector. The above is a robust least squares problem, see El Ghaoui and Lebret [7]. The decision variables are the scalars $y_1$ and $y_2$, and the vector $y_3$. This problem has two quadratic constraints, viz.

$$(y_1, q - Py_3) \in \text{Qcone}, \qquad \left(y_2, \begin{bmatrix} 1 \\ y_3 \end{bmatrix}\right) \in \text{Qcone}. \tag{11}$$

Given $P$ and $q$, the following MATLAB function (`rls.m`) constructs problem (10) in the standard dual form (8). The $A$ matrix will be in transposed orientation, and is hence denoted as `At`.

```
1    % [At,b,c,K] = rls(P,q)
2    % Creates dual standard form for robust least squares problem "Pu=q".
3    function [At,b,c,K] = rls(P,q)
4
5    [m, n] = size(P);
6    % ---------- minimize y_1 + y_2 ----------
7    b = -sparse([1; 1; zeros(n,1)]);
8    % ---------- (y_1, q - P y_3) in Qcone ----------
```

```
9    At = sparse([-1, zeros(1,1+n); ...
10              zeros(m,2), P]);
11   c = [0;q];
12   K.q = [1+m];
13   % ---------- (y_2, (1,y_3)) in Qcone ----------
14   At = [At; 0, -1, zeros(1,n); ...
15              zeros(1,2+n); ...
16              zeros(n,2), -eye(n)];
17   c = sparse([c; 0;1;zeros(n,1)]);
18   K.q = [K.q, 2+n];
```

Notice first that the above function uses sparse data types, in order to save memory. Furthermore, a structure K is defined, with a field K.q that lists the dimensions of the quadratic cones. (The 'q' in K.q stands for 'quadratic'.) The K–structure will be used to tell SeDuMi that the components of $c - A^{\mathrm{T}}y$ are *not* restricted to be nonnegative as they would be in linear programming. Instead, the first K.q(1) entries are restricted to a quadratic cone, and the last K.q(2) entries are restricted to another quadratic cone. This is the way in which we model the symmetric cone $\mathcal{K}$ in (7) and (8), and hence construct the two quadratic constraints in (11).

As a numerical example, we solve a $4 \times 3$ robust least squares problem with dependent columns in $P$. The example is from [7].

```
>> P = [3 1 4;0 1 1;-2 5 3; 1 4 5]; q = [0;2;1;3];
>> [At,b,c,K] = rls(P,q);
>> [x,y,info] = sedumi(At,b,c,K);
SeDuMi by Jos F. Sturm, 1998.
Alg = 1: v-corrector, theta = 0.250, beta = 0.500
eqs m = 5, order n = 5, dim = 11, blocks = 3
 it :     cx        gap   delta  rate  t/maxt   feas
  0 :            5.00E+00 0.000
  1 : -1.23E+01 1.30E+00 0.000 0.2605 0.9000  -0.18
  2 : -5.94E+00 3.34E-01 0.000 0.2568 0.9000   0.54
  3 : -3.60E+00 6.14E-02 0.116 0.1838 0.9000   0.86
  4 : -3.34E+00 1.80E-03 0.000 0.0293 0.9900   1.10
* 5 : -3.33E+00 4.00E-06 0.000 0.0022 0.9990   1.00
* 6 : -3.33E+00 9.59E-09 0.000 0.0024 0.9990   1.00
* 7 : -3.33E+00 6.06E-10 0.153 0.0632 0.9900   1.00
* 8 : -3.33E+00 1.24E-10 0.000 0.2037 0.9000   1.00
iter seconds digits       c*x                b*y
  8      0.1  11.3 -3.3329085968e+00 -3.3329085968e+00
|Ax-b| =   2.8e-16, |x|=  2.0e+00, |y|=  2.5e+00
```

In the above call to SeDuMi, we see a new input argument, viz. K. This argument makes SeDuMi solve an optimization problem in the form (7)–(8), where the symmetric cone $\mathcal{K}$ is described by the structure $\mathcal{K}$. Without the fourth input argument (K), SeDuMi would solve a linear programming problem of the form (1)–(2).

To check that (11) is indeed satisfied by the solution $y$, it is in principle possible to verify the inequality in definition (9) directly. However, it is more convenient to use the function eigK, which is part of SeDuMi. This function returns the eigenvalues (or spectral values) of a vector with respect to a symmetric cone. A symmetric cone consists of those vectors which have nonnegative eigenvalues, see e.g. the book by Faraut and Korányi [9]. For a quadratic cone (9), there are merely two eigenvalues, viz. given a vector $(x_1, x_2) \in \Re \times \Re^{N-1}$, we have $\lambda_1(x_1, x_2) = (x_1 - \|x_2\|)/\sqrt{2}$ and $\lambda_2(x_1, x_2) = (x_1 + \|x_2\|)/\sqrt{2}$.

We can thus check feasibility and optimality as follows:

```
>> [eigK(x,K), eigK(c-At*y,K)]

ans =

    0.0000   -0.0000
    1.4142    3.2307
    0.0000   -0.0000
    1.4142    1.4827

>> x'*(c-At*y)

ans =

   1.5807e-11
```

For symmetric cones $\mathcal{K}$, it holds that $x^{\mathrm{T}} z \geq 0$ for all $x \in \mathcal{K}$ and $z \in \mathcal{K}$. Therefore, $x$ provides an optimality certificate for $y$ just as in the case of linear programming. The interpretation of Farkas dual solutions extends in the same way. See the survey paper of Luo, Sturm and Zhang [16] for the details. However, a paradoxal phenomenon can occur, viz. that $x$ and $y$ are almost feasible, whereas $c^{\mathrm{T}} x - b^{\mathrm{T}} y$ is considerably negative ($\|x\|$ and/or $\|y\|$ must then obviously be very large). SeDuMi will then report an infinite number of digits in accuracy, according to formula (4). This phenomenon was explained by Luo, Sturm and Zhang [15] and Sturm [25].

It is possible that an optimization model has both nonnegativity and quadratic cone constraints. For instance, we may extend the above example with the restriction that $y_3[1] \leq -0.1$, where $y_3[1]$ denotes the first component in the vector $y_3$. This restriction can be added to the model as follows:

```
>> a1 = zeros(1,length(y)); a1(3) = 1;
```

```
>> c = [-0.1; c];  At = [a1;At];
>> K.l = 1;
>> [x,y,info] = sedumi(At,b,c,K);
>> eigK(c-At*y,K)'

ans =

   0.0000   -0.0000    3.2307   -0.0000    1.4904
```

The field `K.l` is the number of nonnegative variables, which in this case is one. (The 'l' in `K.l` stands for 'linear'.) By convention, the nonnegative variables are always the first components, so that $\mathcal{K} = \Re_+ \times$ Qcone $\times$ Qcone in our case. As can be seen from the output of `eigK`, there are 5 eigenvalues for this cone: 1 for each nonnegativity constraint, and 2 for each quadratic constraint. We say that $\mathcal{K}$ is a symmetric cone of order 5. (`SeDuMi` reports 'order $n = 6$', because of its internal self–dual reformulation.)

`SeDuMi` supports an alternative form of the quadratic cone, viz.

$$\text{Rcone} := \left\{ (x_1, x_2, x_3) \in \Re \times \Re \times \Re^{N-2} \,\middle|\, x_1 x_2 \geq \frac{1}{2}\|x_3\|^2,\; x_1 + x_2 \geq 0 \right\}. \tag{12}$$

Geometrically, Rcone is simply a rotation of Qcone. The specific form of Rcone is convenient for modeling convex quadratic functions. Namely, by adding the linear equality constraint '$x_1 = 1$' to the model, we obtain the restriction

$$x_2 \geq \frac{1}{2}\|x_3\|^2.$$

Throughout the model, we can then use $x_2$ as a tight upper bound on $\|x_3\|^2/2$. Fractions are also conveniently modeled by Rcone constraints. For instance, we may minimize $1/x_1$ for $x_1 > 0$ by solving the model

$$\min\{x_2 \mid x_1 x_2 \geq 1,\; x_1 + x_2 \geq 0\}.$$

Notice that this problem does not have a solution: the infimum of $1/x_1$ is zero, for $x_1 \to \infty$.

```
>> clear K;
>> c = [0, 1, 0]; b = sqrt(2); A = [0, 0, 1]; K.r = 3;
>> [x,y,info] = sedumi(A,b,c,K);
>> x(2), x(1)*x(2)

ans =

   1.5360e-05
```

```
ans =

    1.0147
```

You may find that $x_2$ is not yet close enough to zero, and that $x_1$ is not equal to $\infty$ either. However, the primal solution is feasible, the dual solution is almost feasible, and the duality gap is even negative. This illustrates an error bound difficulty, which is usual for this type of irregular problems. In Section 5, we will see how to obtain a more accurate solution, by setting an optional parameter, `pars.eps`.

As illustrated by the above example, the field `K.r` serves to list the dimensions of Rcone constraints, analogously to the definition of Qcone constraints by `K.q`. (The 'r' in `K.r` stands for 'rotated quadratic cone'.) Setting both `K.l`, `K.q` and `K.r` fields yields a symmetric cone of the form
$$\mathcal{K} = \Re_+^{K.l} \times (\text{Qcone} \times \cdots \times \text{Qcone}) \times (\text{Rcone} \times \cdots \times \text{Rcone}).$$

For instance, we can add a bound '$x_1 \leq 10^7$' to the model as follows:

```
>> c = [0, 0, 1, 0]; b = [sqrt(2); 1E7]; A = [0, 0, 0, 1;1, 1, 0, 0];
>> K.l = 1; K.r = 3;
>> [x,y,info] = sedumi(A,b,c,K);
```

Some applications of Qcone and Rcone constraints are discussed in Lobo et al. [14].

## 3.2   The positive semidefinite cone

Semidefiniteness constraints are an important class of restrictions that can be modeled with `SeDuMi`. As an example, consider the following problem:

$$\min\left\{ \sum_{i=1}^{m}(m-i)x_{ii} \,\middle|\, \sum_{i=1}^{m-k} x_{i,i+k} = b_k \text{ for } k = 0,\ldots,m-1, \, X \text{ is psd} \right\}. \tag{13}$$

Here, $X$ is an $m \times m$ symmetric matrix, and $x_{ij}$ denotes the entry on row $i$ and column $j$. The length $m$ vector $b$ is given. The abbreviation 'psd' stands for 'positive semidefinite'. The above optimization problem yields a minimal phase spectral factorization of an autocorrelation vector $b$, see Davidson, Luo and Sturm [4]. Problem (13) is stated in terms of an $m \times m$ symmetric matrix of decision variables, whereas `SeDuMi` works with a vector of decision variables, as in (7)–(8). This small issue is resolved by using the well known technique of vectorization. Vectorization is implemented by the functions `vec` and `mat`, which are part of `SeDuMi`. The function `vec(X)` creates a long vector, by stacking the columns of the matrix $X$, as in:

```
>> x = vec([1, 5, -3;  5, 2, -9;  -3, -9, 4])'

x =

    1    5   -3    5    2   -9   -3   -9    4
```

The inverse of `vec` is `mat`. Thus, if $x$ is a vector of length $n^2$, then $\mathtt{mat}(x)$ constructs an $n \times n$ matrix, and fills it with the entries of the vector $x$, starting at the first column.

```
>> mat(x)

ans =

    1    5   -3
    5    2   -9
   -3   -9    4
```

The following MATLAB function produces a standard primal form for problem (13).

```
1    % [At,b,c,K] = specfac(b)
2    % Creates primal standard form for minimal phase spectral factorization.
3    function [At,b,c,K] = specfac(b)
4
5    m = length(b);
6    % ---------- minimize sum (m-i)*X(i,i) ----------
7    c = vec(spdiags((m-1:-1:0)',0,m,m));
8    % ----- Let e be all-1, and allocate space for the A-matrix -----
9    e = ones(m,1);
10   At = sparse([],[],[],m^2,m,m*(m+1)/2);
11   % ---------- sum(diag(X,k)) = b(k) ----------
12   for k = 1:m
13      At(:,k) = vec(spdiags(e,k-1,m,m));
14   end
15   K.s = [m];
```

The field `K.s` $= [m]$ tells `SeDuMi` that we want the $m \times m$ matrix `mat(x)` to be symmetric positive semidefinite. (The 's' in `K.s` stands for 'semidefinite'.) We can now solve problem (13) as follows:

```
>> b = [2; 0.2; -0.3];
>> [At,b,c,K] = specfac(b);
>> [x,y,info] = sedumi(At,b,c,K);
```

```
SeDuMi by Jos F. Sturm, 1998.
Alg = 1: v-corrector, theta = 0.250, beta = 0.500
eqs m = 3, order n = 4, dim = 10, blocks = 2
 it :     cx        gap  delta  rate  t/maxt   feas
  0 :          4.00E+00 0.000
  1 :  8.14E+00 1.40E+00 0.000 0.3497 0.9000   0.32
  2 :  2.29E+00 4.68E-01 0.000 0.3346 0.9000   0.59
  3 :  3.42E-01 1.12E-01 0.337 0.2391 0.9000   0.84
  4 :  1.26E-01 1.92E-03 0.000 0.0172 0.9900   1.24
* 5 :  1.23E-01 3.97E-06 0.000 0.0021 0.9990   1.00
* 6 :  1.23E-01 8.88E-10 0.000 0.0002 0.9999   1.00
* 7 :  1.23E-01 2.27E-12 0.000 0.0026 0.9990   1.00
iter seconds digits       c*x               b*y
  7      0.1  10.7  1.2273256502e-01  1.2273256502e-01
|Ax-b| =   0.0e+00, |x|=  2.0e+00, |y|=  7.6e-01
```

To check positive semidefiniteness, we can either use the function `eig` that is part of MATLAB, or the function `eigK`, which comes with SeDuMi.

```
>> [eig(mat(x)), eigK(x,K)]

ans =

    0.0000    0.0000
    0.0000    0.0000
    2.0000    2.0000
```

The use of `eigK` is more convenient, especially if there are multiple semidefiniteness constraints, or if there are also nonnegativity or quadratic cone constraints. SeDuMi will always produce symmetric matrix variables, i.e. `mat(x)` is symmetric. Do not add symmetry constraints explicitly, as in '$x_{ij} - x_{ji} = 0$'. At best, such constraints will be removed by SeDuMi from the $A$ matrix.

However, the dual solution $c - A^\mathrm{T} y$ need not be symmetric, as can be seen in the numerical example that we are dealing with:

```
>> mat(c-At*y)

ans =

    2.0727   -0.3130    0.6849
         0    1.0727   -0.3130
         0         0    0.0727
```

In this case, the dual solution is upper triangular, because `mat(c)` is diagonal, and `mat(At(:,k))` is upper triangular for all $k = 1, 2, \dots, m$. Letting $Z = \mathtt{mat}(c - A^{\mathrm{T}}y)$, SeDuMi restricts the symmetric part of $Z$, which is $(Z + Z^{\mathrm{T}})/2$, to be positive semidefinite. The function `eigK` yields the eigenvalues of the symmetric part. Thus,

```
>> eigK(c-At*y,K)
```

```
ans =
```

```
    1.0583
    2.1597
   -0.0000
```

produces the same result as

```
>> Z = mat(c-At*y); eig(Z+Z')/2
```

Notice that problem (13) is equivalent to

$$\min\left\{ \sum_{i=1}^{m}(m-i)x_{ii} \,\middle|\, \sum_{i=1}^{m-k} \frac{x_{i,i+k} + x_{i+k,i}}{2} = b_k \text{ for } k = 0, \dots, m-1,\ X \text{ is psd} \right\}. \tag{14}$$

Namely, $x_{i,i+k} = (x_{i,i+k} + x_{i+k,i})/2$, because $X$ is symmetric. Thus, we may change the $A$ matrix as follows:

```
>> for k=1:size(At,2), Ak = mat(At(:,k)); At(:,k) = vec(Ak+Ak')/2; end
```

The solutions $x$ and $y$, as produced by SeDuMi, will be exactly the same. However, since the constraints in the $A$ matrix have been symmetrized, we find that $\mathtt{mat}(\mathtt{c} - \mathtt{At} * \mathtt{y})$ is now symmetric; it is the matrix $(Z + Z')/2$.

For SeDuMi, it does not make any difference whether the constraints in $A$ and the objective $c$ are symmetrized or not. However, when modeling in the primal standard form, you will probably find it more natural to work with upper or lower triangular matrices in $A$ and $c$; your model will also use less memory like this. On the other hand, symmetric matrices are more natural when modeling in the dual form.

There can be multiple positive semidefiniteness constraints, in which case `K.s` lists the orders of the respective matrices. This is analogous to the definition of multiple quadratic constraints in `K.q` and/or `K.r`. The positive semidefinite variables are always the last components of $x$ and $c - A^{\mathrm{T}}y$, i.e.

$$\mathcal{K} = \Re_{+}^{\mathrm{K.l}} \times (\mathrm{Qcone} \times \cdots \times \mathrm{Qcone}) \times (\mathrm{Rcone} \times \cdots \times \mathrm{Rcone}) \times (\mathrm{Scone} \times \cdots \times \mathrm{Scone}),$$

where Scone denotes the cone of positive semidefinite matrices. It is easy to remember the above arrangement, by noting the alphabetical order of 'l', 'q', 'r' and 's'.

# 4 complex values

In some application areas, such as signal processing, optimization problems may involve complex valued data. An example is the Toeplitz Hermitian covariance estimation problem, which is discussed in Wu, Luo and Wong [30]. Other structured covariance estimation problems, such as discussed in Deng and Hu [5], can be treated similarly. Given a Hermitian matrix $P$, the goal is to find a Hermitian positive definite matrix $Z$ with a Toeplitz structure, such that $\|P - Z\|_F$ is minimal. Thus, the optimization problem is:

$$
\begin{array}{ll}
\text{minimize} & \sum_{i=1}^{m} \left( (z_{ii} - p_{ii})^2 + 2 \sum_{j=i+1}^{m} |z_{ij} - p_{ij}|^2 \right) \\
\text{such that} & Z \text{ is Toeplitz, i.e. } z_{i,j} = z_{i+1,j+1} \text{ for all } i, j = 1, 2, \ldots, m-1 \\
& Z \text{ is psd.}
\end{array}
\tag{15}
$$

If the matrix $P$ has complex entries, then we will usually also see complex entries in the optimal solution $Z$. Notice that the Toeplitz property is better modeled in the dual form, than in the primal form. In fact, `mat(At*y)` in (13) is an upper triangular real Toeplitz matrix, and in (14), it is a symmetric Toeplitz matrix. The MATLAB formulation of (15) therefore resembles the MATLAB formulation of (13).

```
1    % [At,b,c,K] = toepest(P)
2    % Creates dual standard form for Toeplitz-covariance estimation
3    function [At,b,c,K] = toepest(P)
4
5    m = size(P,1);
6    % ---------- maximize y(m+1) ----------
7    b = [sparse(m,1); 1];
8    % ----- Let e be all-1, and allocate space for the A-matrix -----
9    e = ones(m,1);
10   K.q = [1 + m*(m+1)/2];
11   K.xcomplex = 2:K.q(1);        %Norm-bound entries are complex valued
12   At = sparse([],[],[],K.q(1) + m^2,m+1,1 + 2*m^2);
13   % ---------- constraints ----------
14   % -y(m+1) >= norm( vec(P) - sum(y_i * Ti) )      (Qcone)
15   % sum(y_i * Ti) is psd                           (Scone)
16   % --------------------------------
17   At(:,1) = [sparse(2:(m+1),1,1,K.q(1),1); -vec(speye(m))];
18   c = [0; diag(P)];
19   firstk = m+2;
20   for k = 1:(m-1)
21      lastk = firstk + m-k-1;
22      Ti = spdiags(e,k,m,m);
```

```
23    At(:,k+1) = [sqrt(2) * sparse(firstk:lastk,1,1,K.q(1),1); -2*vec(Ti)];
24    c = [c; sqrt(2) * diag(P,k)];
25    firstk = lastk + 1;
26  end
27  At(:,m+1) = [1; sparse(K.q(1) + m^2-1,1)];   % "objective" variable y(m+1)
28  c = [c; zeros(m^2,1)];                       % all-0 in the psd-part
29  K.s = [m];
30  K.scomplex=1;                                %Complex Hermitian PSD
31  % ---------- y(2:m) complex, y(1) and y(m+1) real ----------
32  K.ycomplex = 2:m;
```

We have modeled the objective function by means of an artificial variable, $y_{m+1}$, and $y_{m+1}^2$ is bounded from below by the original quadratic objective function, using a $1 + m(m + 1)/2$-dimensional quadratic cone. The Toeplitz matrix is modeled as

$$y_1 I + 2 \sum_{i=1}^{m-1} y_{i+1} T_i, \tag{16}$$

where $T_i$ is all-1 along the $k$-th upper diagonal, and zero everywhere else. Recall from problem (13) in Section 3.2, that in the real case, SeDuMi restricts the symmetric part of $\mathtt{mat}(c - A^{\mathrm{T}}y)$ to be positive semidefinite. In the complex case, SeDuMi restricts the *Hermitian part*, i.e. $\mathtt{mat}(c - A^{\mathrm{T}}y) + \mathtt{mat}(c - A^{\mathrm{T}}y)'$, to be positive semidefinite. Letting $Z$ denote the Hermitian part of (16), we have

$$Z = y_1 I + \sum_{i=1}^{m-1} (y_{i+1} T_i + \overline{y}_{i+1} T_i^{\mathrm{T}}),$$

where $\overline{y}_{i+1}$ denotes the complex conjugate of $y_{i+1}$. Thus, we have indeed modeled $Z$ as a Hermitian Toeplitz matrix, and SeDuMi further restricts it to be positive semidefinite, because of the field K.s. Furthermore, we tell SeDuMi to allow complex values for $y_2, y_3, \ldots, y_m$, by setting K.ycomplex = 2:m. Remark that unlike K.l, K.q, K.r and K.s, the field K.ycomplex is not involved in the definition of the symmetric cone $\mathcal{K}$ in (7)–(8).

The following lines show how to solve problem (15), for a particular $3 \times 3$ Hermitian matrix $P$, which is neither Toeplitz, nor positive semidefinite.

```
>> i = sqrt(-1);
>> P = [4, 1+2*i, 3-i;  1-2*i, 3.5, 0.8+2.3*i; 3+i, 0.8-2.3*i, 4]

P =

   4.0000             1.0000 + 2.0000i   3.0000 - 1.0000i
   1.0000 - 2.0000i   3.5000             0.8000 + 2.3000i
```

```
   3.0000 + 1.0000i   0.8000 - 2.3000i   4.0000

>> [At,b,c,K] = toepest(P);
>> [x,y,info] = sedumi(At,b,c,K);
>> z = c-At*y; Z = mat(z(K.q+1:length(z))); Z = (Z+Z')/2

Z =

   4.2827                0.8079 + 1.7342i   2.5574 - 0.7938i
   0.8079 - 1.7342i   4.2827                0.8079 + 1.7342i
   2.5574 + 0.7938i   0.8079 - 1.7342i   4.2827

>> eigK(z,K)'

ans =

   -0.0000    2.0517    0.0000    7.2810    5.5670
```

Instead of using the `mat()` function, one may use the `cellK()` function as follows:

```
>> z = cellK(c-At*y,K); Z=z.s{1}; Z=(Z+Z')/2
```

We have found the optimal positive semidefinite Toeplitz matrix $Z$, which has eigenvalues 0, 7.281 and 5.567. Checking the objective values reveals a new phenomenon:

```
>> [c'*x; b'*y]

ans =

  -1.4508 - 0.2428i
  -1.4508
```

The value of $c^{\mathrm{H}}x$, where $^{\mathrm{H}}$ means complex conjugate transpose, may no longer be real, and the same is true for $b^{\mathrm{H}}y$ in general. Obviously, we cannot minimize or maximize complex valued functions. Instead, SeDuMi minimizes Re $c^{\mathrm{H}}x$ in the primal, and maximizes Re $b^{\mathrm{H}}y$ in the dual. Here, Re stands for real part. In the sequel, we will also use the notation Im , to denote the imaginary part.

If we make K.ycomplex = [], then all dual multipliers $y_i$ are restricted to be real.

```
>> K.ycomplex = [];
>> [x2,y2,info2]=sedumi(At,b,c,K);
>> [c'*x2; b'*y2]
```

```
ans =

  -4.5592 - 0.3816i
  -4.5592
```

Clearly, by restricting $y$ to be real, the dual optimal value Re $b^{\mathrm{H}}y = -y_{m+1}$ gets worse. Apparently, something has changed in the primal problem as well, since the primal optimal value has improved from $-1.4508$ to $-4.5592$. The difference is in the '$Ax = b$' restriction, as the following lines show:

```
>> [b-At'*x b-At'*x2]

ans =

   0.0000             -0.0000
   0.0000 + 0.0000i   -0.0000 + 1.8863i
  -0.0000             -0.0000 - 0.4387i
        0                   0
```

The restriction '$Ax = b$' is interpreted by `SeDuMi` as

$$
\begin{cases}
a_i^{\mathrm{H}}x = b_i \text{ if } i \in \texttt{K.ycomplex} \\
\text{Re } a_i^{\mathrm{H}}x = b_i \text{ otherwise.}
\end{cases}
\tag{17}
$$

By making `K.ycomplex = []`, we therefore removed the restrictions on Im $Ax$, and implicitly added the restriction that Im $y = 0$. Complex $y$-variables in the dual form correspond with complex equality constraints in the primal form.

If `size(A,2) = length(b)`, then the primal feasibility requirements are `A'*x = b`, using the complex conjugate transpose $A^{\mathrm{H}}$.

The field `K.scomplex` contains a list of the PSD matrix variables, of order `K.s(K.scomplex)`, which are restricted to be Hermitian positive semidefinite matrices. For the remaining matrix variables, the primal $x$-variables are restricted to be real symmetric positive semidefinite, whereas the dual slack variables are restricted to be positive semi-definite on the real part only (the dual imaginary part is then unrestricted).

The field `K.xcomplex` lists the primal $x$-variables that are allowed to have a nonzero imaginary part. For the free and nonnegative $x$-components, ths imaginary part is then unrestricted in sign. For example, the restriction '$x_i \geq 0$' is interpreted by `SeDuMi` as

$$
\begin{cases}
x_i \in \Re_+ \text{ if } i \notin \texttt{K.xcomplex} \\
\text{Re } x_i \geq 0 \text{ if } i \in \texttt{K.xcomplex.}
\end{cases}
\tag{18}
$$

A similar convention holds for the first entry in a q-second order cone. The remaining entries in a second order cone that are listed in `K.xcomplex` are simply complex variables that appear in the norm-bound restriction. Only entries in the `f,l,q,r`-cones can be listed in `K.xcomplex`; the matrix variables are handled by the field `K.scomplex`.

On the dual side, `K.xcomplex` lists the equality and nonnegativity constraints for which the restriction $\mathrm{Im}\, c_i - a_i^{\mathrm{T}} y = 0$ must be imposed on the imaginary part. This interpretation also works for the first entry in a second order q-cone. The remaining entries in a second order cone that are listed in `K.xcomplex` are simply complex variables that appear in the norm-bound restriction (this is completely symmetric to the primal).

For sensitivity analysis, it is interesting to note that $\mathrm{Re}\,(\Delta c)^{\mathrm{H}} x$ is a supergradient for the optimal value function, under perturbations of the form $c + t\Delta c$, whereas $\mathrm{Re}\,(\Delta b)^{\mathrm{H}} y$ is a subgradient of the optimal value function under perturbation of $b$. For a discussion of sensitivity analysis in (real symmetric) semidefinite programming, see Goldfarb and Scheinberg [12].

## 5 optional settings

By default, `SeDuMi` fills your terminal screen with some output concerning its iterative progress. This can be an annoying feature, in particular if `SeDuMi` is merely used as a subroutine within a larger program. To suppress the on-screen output of `SeDuMi`, it suffices to set an optional parameter, `pars.fid`, to zero.

```
>> load truss1
>> pars.fid = 0;
>> [x,y,info] = sedumi(At,b,c,K,pars);
```

The structure `pars` is not only used for suppressing iterative output of `SeDuMi`. It can contain a number of optional fields, which we will discuss in this section.

The abbreviation 'fid' in `pars.fid` stands for 'file identifier': the output of `SeDuMi` will be sent to the file whose file identifier is `pars.fid`. The file identifier for the null-device is 0, which is useful for suppressing output, and for the terminal screen it is 1. Output can also be redirected to a file, e.g.

```
>> pars.fid = fopen('truss1.out','w');
>> [x,y,info]=sedumi(At,b,c,K,pars);
>> fclose(pars.fid); pars.fid = 1;
```

With the above lines, the output is redirected to the file 'truss1.out', as can be checked with the command `dbtype truss1.out`.

`SeDuMi` uses a variant of the primal–dual interior point method, which is known as the centering–predictor–corrector method [25]. There are 3 variants of the centering–predictor–corrector method implemented, which can be selected with the field `pars.alg`. With `pars.alg`

Figure 1: Plot produced by setting `pars.vplot = 1`.

= 0, you select a longest-step algorithm, without any second order corrector. To enhance the algorithm with a second order corrector, you can either set `pars.alg = 1` or `pars.alg = 2`. With `pars.alg = 1`, the second order corrector is derived by linearization of the so-called $v$-values, whereas `pars.alg = 2` uses linearization of the squared $v$-values, which is also known as $xz$-linearization. For linear programming, $xz$-linearization results in the well-known Mehrotra's corrector [17]. In all three variants, the centering step is determined by the central region parameter, `pars.theta`. This parameter can take any value in $(0, 1]$. At one extreme, `pars.theta = 1` results in path–following, which typically involves relatively short step lengths. Setting `pars.theta` to a smaller value, such as $1/4$, makes the algorithm work in the neighborhood of a full dimensional central region, and this typically allows for larger step lengths, see Sturm and Zhang [26]. The size of the neighborhood is controled by the parameter `pars.beta`, which can be assigned any value in $(0, 1)$. In the output of `SeDuMi` on the terminal screen, there is a column labeled 'delta', which lists the actual distance to the central region in each iteration. The step length will always be such that this is at most `pars.beta`. The ratio of the actual step length and the maximal steplength to the boundary of the cone $\mathcal{K}$ is listed in the column labeled 't/maxt'. For some iterations, an asterisk ('*') appears in front of the output line. At these iterations, the residual vector of the self–dual model has been recomputed (to avoid accumulation of numerical errors).

For research purposes, `SeDuMi` can produce a plot of the iterative $v$–values. This feature is activated by setting `pars.vplot = 1`. For problem `truss1`, this results in the plots of Figure 1. For each iteration, the first plot shows all the $v$–values, divided by the mean of the $v$-vector in that iteration. It also gives a horizontal line at value 1, representing the central path, and a horizontal line at the central region threshold, `pars.theta = 1/4`. Any $v$-values below this

threshold will be corrected by the centering component in the succeeding iteration. The second plot shows the rate of linear reduction, which is simply

$$\frac{\text{duality gap in iteration } k}{\text{duality gap in iteration } k-1}.$$

The rate of linear reduction is also listed in the column 'rate' in the on-screen output of `SeDuMi`, and the iterative duality gap is listed under 'gap'. This is the duality gap in an artificial self–dual model, in which your original model is embedded by `SeDuMi`, using the technique of Ye, Todd and Mizuno [31]. The self–dual model gives rise to a feasibility indicator, listed in the column 'feas'. Ideally, the indicator converges to $+1$ for feasible problems, and to $-1$ for (primal and/or dual) infeasible problems.

Termination control is provided by the fields `maxiter`, and `eps` in the `pars` structure. `SeDuMi` will terminate successfully if it finds a solution that violates feasibility and optimality requirements by no more than `pars.eps`. The parameter `pars.maxiter` allows you to set a maximum on the number of iterations. By default, `pars.eps = 1E-9` and `pars.maxiter = 150`. A possible experiment with these parameters is to set `pars.eps = 0` in the example of minimizing $1/x_1$, which was discussed in Section 3.1.

**Acknowledgments.** I thank T. Terlaky for encouraging me to write this manual, and for pointing out a bug in the first public release of `SeDuMi`. P. Apkarian, M. Bengtsson, T.N. Davidson, F. Glineur, V. Prodanovic and A. Ross helped to improve the software, by providing bug reports and suggestions on the first release of the software. Numerous other users have contributed in the same manner since. Two anonymous referees have contributed in improving this document.

## Footnotes

*MATLAB is a registered trademark of The MathWorks, Inc.

†Appeared in: Optimization Methods and Software 11–12 (1999) 625–653

‡Research up to version 1.02 performed at Communications Research Laboratory, McMaster University, Hamilton, Canada. Supported by Netherlands Organization for Scientific Research (NWO).

[1]This function has not been updated since Version 1.00, and may not be compatible with the latest software

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

# A    Calling Sequence

The primal canonical form for solving optimization problems with SeDuMi is

$$\min\{c^{\mathrm{T}}x \mid Ax = b, \ x \in \mathcal{K}\},$$

and the dual canonical form is

$$\max\{b^{\mathrm{T}}y \mid c - A^{\mathrm{T}}y \in \mathcal{K}\}.$$

The general calling sequence for solving the above primal–dual pair is

```
[x,y,info] = sedumi(A,b,c,K,pars)
```

Here, `K` is a MATLAB structure to define the symmetric cone $\mathcal{K}$; it consists of the following (optional) fields:

`K.f` The number of free primal variables, i.e. the number of dual equality constraints

`K.l` The number of nonnegativity constraints

`K.q` A list of dimensions of quadratic cone constraints

`K.r` A list of dimensions of rotated quadratic cone constraints

`K.xcomplex` A list of primal variables in the `f,l,q,r` blocks that are allowed to have a nonzero imaginary part. The imaginary parts of the associated dual constraints are then explicitly restricted.

`K.s` A list of orders of positive semidefiniteness constraints

`K.scomplex` A list of matrix variables that are restricted to be Hermitian positive semidefinite.

`K.ycomplex` This field is not related to $\mathcal{K}$. It lists the components of the $y$-variables that are complex valued. Equivalently, it lists the primal equality constraints $(Ax)_i = b_i$ that have to be satisfied not only in their real parts, but also in their imaginary parts.

The structure `K` defines $\mathcal{K}$ to be

$$\mathcal{K} = \Re^{\text{K.f}} \times \Re_+^{\text{K.l}} \times (\text{Qcone} \times \cdots \times \text{Qcone}) \times (\text{Rcone} \times \cdots \times \text{Rcone}) \times (\text{Scone} \times \cdots \times \text{Scone}),$$

In total, the number of Qcone (Rcone, Scone) components is `length(K.q)` (`length(K.r)`, `length(K.s)`). The $i$th Qcone is

$$\text{Qcone}_i = \{(x_1, x_2) \in \Re \times \mathcal{C}^{\text{K.q[i]}-1} \mid x_1 \geq \|x_2\|\},$$

where $\mathcal{C}^n$ denotes the space of complex $n$–tuples. The $j$th Rcone is

$$\text{Rcone}_j = \left\{ (x_1, x_2, x_3) \in \Re \times \Re \times \mathcal{C}^{\text{K.r[j]}-2} \left| x_1 x_2 \geq \frac{1}{2}\|x_3\|^2, \; x_1 + x_2 \geq 0 \right. \right\},$$

and the $k$th Scone is

$$\text{Scone}_k = \left\{ x \in \mathcal{C}^{\text{K.s[k]}^2} \mid \text{mat}(\text{x}) \text{ is Hermitian positive semidefinite} \right\},$$

for primal components. For dual components $z = c - A^{\text{T}} y$, we use a slightly milder definition of Scone, viz.

$$\text{Scone}_k = \left\{ z \in \mathcal{C}^{\text{K.s[k]}^2} \mid \text{mat}(\text{z}) + \text{mat}(\text{z})' \text{ is positive semidefinite} \right\}.$$

If $k$ is *not* listed in `K.scomplex`, then the definition in the dual is even milder, namely,

$$\text{Scone}_k = \left\{ z \in \mathcal{C}^{\text{K.s[k]}^2} \mid \text{Re} \, \text{mat}(\text{z}) + \text{mat}(\text{z})' \text{ is positive semidefinite} \right\}.$$

The length of the vector $c$ should be

```
length(c) = K.f + K.l + sum(K.q) + sum(K.r) + sum(K.s.^2)
```

If the data $(A, b, c)$ is real valued, then $x$ and $y$ will also be real valued.

The parameter `pars` is a MATLAB structure, consisting of the following (optional) fields:

`pars.fid` By default, `pars.fid=1`, which tells SeDuMi to produce iterative statistics on the screen. If `pars.fid=0`, then SeDuMi runs quietly, i.e. no screen output. In general, output is written to the file or device that is identified by the file handle `pars.fid`. A file handle is assigned to a file by the MATLAB function `fopen`, as in

```
pars.fid = fopen('truss1.out','w').
```

**pars.alg** By default, `pars.alg=2`. If `pars.alg=0`, then a first-order algorithm is used, which is not recommendable. If `pars.alg=1`, then `SeDuMi` uses the centering-predictor-corrector algorithm with $v$-linearization. If `pars.alg=2` then $xz$-linearization is used in the corrector, similar to Mehrotra's algorithm. All 3 algorithms are special instances of the generic wide-region algorithm, as discussed in Chapter 7 of Sturm [25].

**pars.theta, pars.beta** By default, `pars.theta=0.25` and `pars.beta=0.5`. These are the wide region and neighborhood parameters. Valid choices are $0 < \theta <= 1$ and $0 < \beta < 1$.

**pars.stepdif, pars.w** By default, `pars.stepdif=1` and `pars.w = [1 1]`. This means that primal/dual step length differentiation is enabled (disabled if `pars.stepdif=0`). The priorities of the relative primal, dual and gap residuals are weighted as `w(1):w(2):1`, in order to find the optimal step differentiation.

**pars.vplot** If this field is 1, then SeDuMi produces a fancy $v$-plot, for research purposes. Default: `vplot = 0`.

**pars.eps** The desired accuracy.

**pars.bigeps** The required accuracy to get `info.numerr < 2`.

**pars.maxiter** Maximum number of iterations, before termination.

**pars.denq** Proportion of x(i)'s for which the sparsity in A(:,i) is considered normal. Default: 0.75.

**pars.denf** A column is treated as dense if it has `pars.denf` times more nonzeros than normal. Default: 10.

**pars.stopat** Enters MATLAB debugging mode at the beginning of iteration `pars.stopat`. Default: -1.

**pars.cg** Various parameters for controlling the Preconditioned conjugate gradient method (CG), which is only used if results from Cholesky are inaccurate. Type 'help sedumi' for details.

**pars.chol** Various parameters for controling the Cholesky solve. Type 'help sedumi' for details.

The output parameter `info` is a MATLAB structure, with the following fields:

**info.pinf** and **info.dinf** The feasibility status of the primal-dual problem pair, as detected by `SeDuMi`. There are three cases:

    1. `pinf = dinf = 0` Then $x$ and $y$ are (approximate) optimal solutions, i.e. $Ax = b$, $x \in \mathcal{K}$, $c - A^{\mathrm{T}}y \in \mathcal{K}$, and $c^{\mathrm{T}}x \le b^{\mathrm{T}}y$ (approximately).

2. `pinf = 1` Primal is infeasible, i.e. $\{x \in \mathcal{K} \mid Ax = b\} = \emptyset$. Then $y$ is a Farkas-type solution, i.e. $b^{\mathrm{T}}y > 0$ and $-A^{\mathrm{T}}y \in \mathcal{K}$.

3. `dinf = 1` Dual is infeasible, i.e. $\{y \mid c - A^{\mathrm{T}}y \in \mathcal{K}\} = \emptyset\}$. Then $x$ is a Farkas-type solution, i.e. $c^{\mathrm{T}}x < 0$, $Ax = 0$ and $x \in \mathcal{K}$.

`info.numerr` A positive value of `info.numerr` means that `SeDuMi` terminated without achieving the desired accuracy, because of numerical problems. If `info.numerr = 1` then the results are merely inaccurate: the solution has still achieved the accuracy denoted by `pars.bigeps`, which is 1E-3 by default. If `info.numerr = 2` then `SeDuMi` failed completely.

[Supplementary Material 3 · SeDuMi_Guide_11.pdf]

# ADDENDUM TO THE SEDUMI USER GUIDE VERSION 1.1

IMRE PÓLIK

## 1. INTRODUCTION

The main goal of this reference guide is to give a summary of all the options in SeDuMi. The default value of the options is satisfactory for general use. If you experience difficulties solving some problems please report them at the SeDuMi Forum (`http://sedumi.mcmaster.ca`). If you need a longer description of SeDuMi and examples of use then consult the old (1.05R5) manual [3].

## 2. INPUT FORMAT

A full-featured SeDuMi call in Matlab is

`>> [x,y,info] = sedumi(A,b,c,K,pars);`

where most of the parts can be omitted. With this call SeDuMi solves the following primal-dual optimization problem:

$$\min c^T x \qquad\qquad \max b^T y$$
$$Ax = b \qquad\qquad A^T y + s = c$$
$$x \in K \qquad\qquad s \in K^*,$$

where $A \in \mathbb{R}^{m \times n}$, $x, s, c \in \mathbb{R}^n$, $y, b \in \mathbb{R}^m$, $K$ is a cone and $K^*$ is its dual cone. Omitting K implies that all the variables are nonnegative. $A$, $b$ and $c$ contain the problem data, if either $b$ or $c$ is missing then it is treated as zero.

The structure $K$ defines the cone in the following way: it can have fields `K.f`, `K.l`, `K.q`, `K.r` and `K.s`, for Free, Linear, Quadratic, Rotated quadratic and Semi-definite. In addition, there are fields `K.xcomplex`, `K.scomplex` and `K.ycomplex` for complex-variables.

(1) `K.f` is the number of FREE, i.e., UNRESTRICTED primal components. The dual components are restricted to be zero. E.g. if `K.f = 2` then `x(1:2)` is unrestricted, and `s(1:2)=0`. These are ALWAYS the first components in `x`.

(2) `K.l` is the number of NONNEGATIVE components. E.g., if `K.f=2`, `K.l=8` then `x(3:10)>=0`.

(3) `K.q` lists the dimensions of LORENTZ (quadratic, second-order cone) constraints. E.g., if `K.l=10` and `K.q = [3 7]` then `x(11) >= norm(x(12:13))`, `x(14)>= norm(x(15:20))`. These components ALWAYS immediately follow the `K.l` nonnegative ones. If the entries in $A$ and/or $c$ are COMPLEX, then the $x$-components in

---

*Date*: June 16, 2005.

Based on the original SeDuMi User Guide and the contents of the SeDuMi help.

`norm(x(#,#))` take complex-values, whenever that is beneficial. Use `K.ycomplex` to impose constraints on the imaginary part of `A*x`.

(4) `K.r` lists the dimensions of Rotated LORENTZ constraints. E.g., if `K.l=10`, `K.q=[3,7]` and `K.r = [4 6]`, then `2*x(21)x(22) >= norm(x(23:24))^2` and `2*x(25)x(26)>=norm(x(27:30))^2`. These components ALWAYS immediately follow the `K.q` ones. Just as for the `K.q`-variables, the variables in `norm(x(#,#))` are allowed to be complex, if you provide complex data. Use `K.ycomplex` to impose constraints on the imaginary part of `A*x`.

(5) `K.s` lists the dimensions of POSITIVE SEMI-DEFINITE (PSD) constraints. If `K.l=10`, `K.q = [3 7]` and `K.s = [4 3]`, then `mat( x(21:36),4 )` is PSD, `mat( x(37:45),3 )` is PSD. These components are ALWAYS the last entries in x.

- `K.xcomplex` lists the components in f,l,q,r blocks that are allowed to have nonzero imaginary part in the primal.
- `K.scomplex` lists the PSD blocks that are Hermitian rather than real symmetric.
- Use `K.ycomplex` to impose constraints on the imaginary part of `A*x`.

The dual multipliers $y$ have analogous meaning as in the `x>=0` case, except that instead of `c-A'*y>=0` resp. `-A'*y>=0`, one should read that `c-A'*y` resp. `-A'*y` are in the cone that is described by `K.l`, `K.q` and `K.s`. In the above example, if `z = c-A'*y` and `mat(z(21:36),4)` is not symmetric/Hermitian, then positive semi-definiteness reflects the symmetric/Hermitian parts, i.e. `s + s'` is PSD.

If the model contains COMPLEX data, then you may provide a list `K.ycomplex`, with the following meaning:

- `y(i)` is complex if `ismember(i,K.ycomplex)`
- `y(i)` is real otherwise

The equality constraints in the primal are then as follows:

- `A(i,:)*x = b(i)` if `imag(b(i)) ~= 0` or `ismember(i,K.ycomplex)`
- `real(A(i,:)*x) = b(i)` otherwise.

Thus, equality constraints on both the real and imaginary part of `A(i,:)*x` should be listed in the field `K.ycomplex`.

You may use `EIGK(x,K)` and `EIGK(c-A'*y,K)` to check that `x` and `c-A'*y` are in the cone `K`.

## 3. Options

Most of the options in SeDuMi can be overridden by specifying them explicitly in the structure `pars`. This structure can have the following fields (the default value is in brackets).

### 3.1. General options.

**pars.fid (1):** The output of SeDuMi is written to the file with handle `fid`. If `fid=0`, then SeDuMi runs quietly, i.e., there is no screen output. Use `fopen` to assign a handle to a file.

**pars.maxiter (150):** The maximum number of iterations. In most cases SeDuMi stops after 20-30 iterations, and almost never needs more than 100 iterations.

**pars.eps** ($10^{-8}$)**:** Desired accuracy. If this accuracy is achieved then `info.numerr` is set to 0. If `pars.eps=0` then SeDuMi runs as long as it can make progress.

**pars.bigeps** ($10^{-3}$)**:** In case the desired accuracy `pars.eps` cannot be achieved, the solution is tagged as `info.numerr=1` if it is accurate to `pars.bigeps`, otherwise it yields `info.numerr=2`.

**pars.alg (2):** If `pars.alg=0`, then a first-order wide region algorithm is used, not recommended. If `pars.alg=1`, then SeDuMi uses the centering-predictor-corrector algorithm with v-linearization. If `pars.alg=2`, then $xz$-linearization is used in the corrector, similar to Mehrotra's algorithm. The wide-region centering predictor-corrector algorithm was proposed in Chapter 7 of [2].

**pars.theta (0.25), pars.beta (0.5):** These are the wide region and neighborhood parameters. Valid choices are $0 <$ `pars.theta` $\leq 1$ and $0 <$ `pars.beta` $< 1$. Setting `pars.theta=1` would restrict the iterates to follow the central path in an $N_2(\beta)$-neighbourhood. In practice, SeDuMi rounds `pars.beta` to be between 0.1 and 0.9 and `pars.theta` to be between 0.01 and 1.

**pars.stepdif (2):** If `pars.stepdif=0` then the primal-dual step differentiation is always disabled, while if `pars.stepdif=1` then it is always enabled. Using step differentiation helps if the problem is ill-conditioned or the algorithm is close to convergence. Setting `pars.stepdif=2` uses an adaptive scheme: it starts SeDuMi with step-differentiation disabled and enables it
  - after 20 iterations, or
  - if `feasratio` is between 0.9 and 1.1, or
  - more than one conjugate gradient iterate is needed.

**pars.w** $(1,1)$**:** If step-differentiation is enabled, SeDuMi weights the relative primal, dual and gap residuals as w(1):w(2):1 in order to find the optimal step differentiation. These number should be greater than $10^{-8}$.

**pars.numtol** $(5 \times 10^{-7})$**, pars.bignumtol** $(0.9)$**, pars.numlvl (0):** These options control some numerical behaviour, don't tamper with them.

## 3.2. Options controlling the preprocessing.

**pars.denf (10), pars.denq (0.75):** Parameters used in deciding which columns are dense and sparse.

**pars.free (1):** Specifies how SeDuMi handles free variables. If `pars.free=0` then free variables are converted into the difference of two nonnegative variables. This method is numerically unstable and unnecessarily increases the number of variables. If `pars.free=1` then free variables are placed inside a Lorentz cone whose first variable is unconstrained. This method is more stable and has also been suggested by Jos Sturm in [4]. Please note however, that this makes the problem nonlinear and it might take longer to solve it. If you experience such behaviour then change this setting to 0.

**pars.sdp (1):** Enables the SDP preprocessing routines, such as detecting diagonal SDP blocks.

## 3.3. Options controlling the Cholesky factorization. These options are subfields of `pars.chol`.

**pars.chol.skip (1):** Enables skipping bad pivots.

**pars.chol.canceltol** ($10^{-12}$)**:** Relative tolerance for detecting cancellation during Cholesky factorization.

**pars.chol.abstol** ($10^{-20}$)**:** Skips pivots falling below this value.

**pars.chol.maxuden (500):** Pivots in dense-column factorization so that these factors satisfy $\max(\text{abs}(\text{Lk})) \leq$ maxuden.

3.4. **Options controlling the Conjugate Gradient refinement.** Various parameters for controlling the Preconditioned conjugate gradient method (CG), which is only used if results from Cholesky factorization are inaccurate.

**pars.cg.maxiter (49):** Maximum number of CG-iterates (per solve). Theoretically needed is `|add|+2*|skip|`, the number of added and skipped pivots in Cholesky.

**pars.cg.refine (1):** Number of refinement loops that are allowed. The maximum number of actual CG-steps will thus be `1+(1+pars.cg.refine)*pars.cg.maxiter`.

**pars.cg.stagtol** ($5 \times 10^{-14}$)**:** Terminates if relative function progress is less than this number.

**pars.cg.restol** ($5 \times 10^{-3}$)**:** Terminates if residual is a `pars.cg.restol` fraction of the duality gap. Should be smaller than 1 in order to make progress.

**pars.cg.qprec (1):** Stores CG-iterates in quadruple precision if `pars.cg.qprec=1`.

3.5. **Debugging options.**

**pars.vplot (0):** If this field is 1, then SeDuMi produces a plot for research purposes.

**pars.stopat (-1):** SeDuMi enters debugging mode at the iterations specified in this vector.

**pars.errors (0):** If `pars.errors=1` then SeDuMi outputs some error measures as outlined at the 7th DIMACS challenge. The errors are both displayed in the output and returned in `info.err`.

## 4. Output format

4.1. **Description of the output fields.** When calling SeDuMi with three output variables (`[x,y,info]=sedumi(...)`) some useful information is returned in the structure `info`. It has the following fields:

**info.cpusec:** Total CPU (not wall clock) time in seconds spent in optimization.

**info.timing:** Detailed CPU timing information in seconds. The three numbers give the time spent in preprocessing, IPM iterations and postprocessing, respectively. Although in most cases IPM iterations take more than 90% of the time, in some cases it can be as low as 50%.

**info.feasratio:** The final value of the feasibility indicator. This indicator converges to 1 for problems with a complementary solution, and to -1 for strongly infeasible problems. If `info.feasratio` is somewhere in between, and the problem is not purely linear then the problem may be nasty (e.g., the optimum is not attained). Otherwise, if the problem is linear then the reason must lie in numerical problems: try to rescale the problem.

**info.pinf, info.dinf:** If `info.pinf=1` then there cannot be an $x \geq 0$ with $Ax = b$, and this is certified by $y$, viz. $b^T y > 0$ and $A^T y \leq 0$ thus $y$ is a Farkas solution. On the other hand if `info.dinf=1` then there cannot be a $y$ such that $c - A^T y >= 0$, and this is certified by $x$, viz. $c^T x < 0$, $Ax = 0$, $x \geq 0$. Thus $x$ is a Farkas solution. If both `info.pinf` and `info.dinf` are 0 then the solution given by SeDuMi is both near primal and dual feasible.

**info.numerr:** Indicates how accurate the solution is. If `info.numerr = 0` then the desired accuracy (specified by `pars.eps`) is achieved. If `info.numerr = 1` then only reduced accuracy (given by `pars.bigeps`) is achieved. Finally, if `info.numerr = 2` indicates complete failure due to numerical problems.

**info.err:** If the option `pars.errors` is set to 1 then SeDuMi outputs some error measures as described in [1].

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

MCMASTER UNIVERSITY, ADVANCED OPTIMIZATION LAB
*E-mail address*: `poliki@mcmaster.ca`