[Reviews · NeurIPS 2020]

Review 1

Summary and Contributions: The paper is concerned with estimating an optimal transport map given samples from the source and target distributions. The idea is to represent source and target marginals, as well as the (bivariate) cost function, as empirical kernel mean embeddings. Actually, a sort of minimum norm like embedding is used for the latter (cost function). Given these assumptions it is shown that the optimal transport map enjoys a representer theorem, meaning it lies in the span of the data. They prove this representer theorem and also an intriguing result that their estimator converges independently of the source and target dimensionality. After the rebuttal: I thank the authors for their response. I maintain my positive appraisal.

Strengths: I really enjoyed the creative aspect here. It's a fun idea and seems to work really well in practice. The authors combine this creative idea with a nice theoretical guarantee.

Weaknesses: The paper really needs a bit of work from an English and overall presentation standpoint including formatting of the figures. This is nothing deal breaking, just that it gives a slightly weak impression on a superficial level. It would be nice if the paper came with a primer on kernel mean embeddings which helped the reader to undewrstand the notation in section 3.

Correctness: I have not checked the detailed proofs. The reviewer load is high these days, I apologise.

Clarity: It's moderately well written. See the weaknesses section.

Relation to Prior Work: Yes, the links are clearly drawn out.

Reproducibility: Yes

Additional Feedback: See the weaknesses section. Also, I find line 151 slightly odd, in the definition of rho star. This is not a minimum norm interpolant. Moreover you are defining rho in terms of the entire function c. Can you expand on how you compute rho star?


Review 2

Summary and Contributions: Rebuttal read. It could be good indeed to put the complexity matters in the main paper. =============================================== The proposed methods consists in re-writing the original optimal transport formulation in terms of kernel embedding, such that the cost function is exactly represented (or arbitrarily close to the true one). The obtained problem then benefits from kernel framework. To estimate the marginal kernel mean embedding, the author propose a regularization based on MMD distances. Eventually the authors obtain a (nice) fully regularized and kernelized problem to represent the original one. Known algorithms are proposed to solve the problems, it is not a major contribution of the paper but it shows that one can find solutions quite efficiently. The experimental part shows that the proposed method can outperform estimates based on discrete OT, using cases for which the analytical solutions are known. It also illustrates the capacity to provide estimation on out-of-sample points. It also proposes a successful application to domain adaptation.

Strengths: The main strength of the paper can be listed as follows: - representer theorem - can be applied to all variants of OT: continuous, semi-discrete, and discrete - the proposed method allows for out-of-sample extension

Weaknesses: I can't see many weaknesses, apart from the complexity matters : I would be curious to have some insights on this subject.

Correctness: I could not find flows in the process of kernelization.

Clarity: yes

Relation to Prior Work: yes

Reproducibility: Yes

Additional Feedback: I found that some parts of the paper really "compressed", I guess due to space constraints. For instance, may be because I'm not much familiar with the task, I found hard to understand the domain adaptation experimental setup. Sentences like "OOS estimation is especially attractive..." (l293) could leave space to some more informative ones?


Review 3

Summary and Contributions: This paper proposed to utilize the kernel embedding method to reduce the dimensionality of the statistical optimal transport (OT) problem. By reformulating the OT problem as a kernel embedding problem, the paper claims that the sample complexity of the proposed estimator is dimension-free under mild conditions.

Strengths: Dimension reduction is an important issue in statistical OT problems. Existing methods mainly focused on regularized approaches and random projection methods. The kernel embedding method, proposed in this paper, can be an interesting complement to existing literature.

Weaknesses: Some implementation details of the proposed method are not clearly specified. The validity of the theoretical claims is hard to check due to the poorly organized proofs. The numerical experiments are not comprehensive. No comparison with popular competitors.

Correctness: (a) The proposed kernel embedding formulation of OT is based on the cross-covariance operator which ignores higher moments distributions. (b) The dimensionality of canonical feature maps can be infinity. This paper did not carefully discuss this issue. Does the proposed method need to make a finite-dimensional approximation? If so, how to selec the number of terms? Does it depend on the dimensionality of the data? (c) The objective functions in (3) and (4) are based on the expectation of functions defined in line 104 which are not empirically available. The algorithm discussed in Section 3.4 is based on a couple of simplification conditions. It will be helpful to discuss the approximation error as well as the computational complexity of the algorithm. (d) The experiment only simulated a multivariate Gaussian OT problem which is in favor of the second-order moment construction as pointed out in (a). It may not be sufficient to demonstrate the effectiveness of the proposed method under general cases.

Clarity: The paper reads smoothly with a couple of grammar errors. The proof in the supplemental material, though not mandatory, are poorly organized and have several gaps. It impedes the reviewer (at least for me) to check the validity of the theory within the time-imited review process. Some details in the experiments are also missing. For example, how to construct \Sigma_1 and \Sigma_2 in the multivariate Gaussian example?

Relation to Prior Work: The paper did not make a comparison with popular dimension reduction methods for statistical OT problems.

Reproducibility: No

Additional Feedback: ######## EDIT after author's response ############# The responses have addressed some of my confusing points. I would like to raise my score to 6. I have not checked the correctness of the theory due to the proofs are not well organized in the appendix.

[Author Response · NeurIPS 2020]

We thank the reviewers for their valuable feedback. We will incorporate the suggestions on the paper write-up and
organization in the final version.

**Reviewer 1**

**Q1)** "Also, I find line 151 slightly odd, in the definition of $\rho^*$. This is not a minimum norm interpolant. Moreover you
are defining $\rho$ in terms of the entire function $c$. Can you expand on how you compute $\rho^*$?"
**A1)** This is the standard definition based on least square interpolation. The formula for $\rho^*$ is $\rho^* = G^{-1}C$, where $G$ is
the gram matrix of mapped source, target data in $H_1 \otimes H_2$ and $C$ is the matrix of cost evaluations at the given source,
target data points. We shall clarify this appropriately in the final draft.

**Reviewer 2**

**Q2)** "I can't see many weaknesses, apart from the complexity matters : I would be curious to have some insights on this
subject."
**A2)** We discuss the computational complexity in the supplementary material (lines $239 - 243$ and $246 - 247$). We
shall include it in the main paper's final version.

**Reviewer 3**

**Q3)** "This paper proposed to utilize the kernel embedding method to reduce the dimensionality of the statistical optimal
transport (OT) problem."
**A3)** This work does not propose to perform dimensionality reduction. It explores the novel idea of posing the OT
problem as that of learning the kernel mean embedding of the optimal transport plan/map from the given samples.

**Q4)** "The proposed kernel embedding formulation of OT is based on the cross-covariance operator which ignores higher
moments distributions."
**A4)** No, the higher order moments are not ignored. Unlike cross-covariance, which is a number, what we use here is the
(kernelized) cross-covariance operator between two RKHS. Infact, whenever moments or expectations exist, all of them
can be calculated from the operator using formula (3.16) in [23]. Instantiations of this formula are also used in our
paper at lines 104, 119 etc.

**Q5)** "The dimensionality of canonical feature maps can be infinity. This paper did not carefully discuss this issue".
**A5)** Indeed the dimensionality of RKHS is infinite. However, thanks to Theorem 2, the fact is that the sample complexity
is still finite. Moreover, the representer theorem (Theorem 3) guarantees finite parameterization for the optimal solution.
To summarize, the issue of the infinite dimensionality of the RKHS is dealt with thoroughly via Theorems 2 and 3.

**Q6)** "Does the proposed method need to make a finite-dimensional approximation? If so, how to selec the number of
terms? Does it depend on the dimensionality of the data?"
**A6)** Nowhere in the paper we make a finite-dimensional approximation nor do we perform dimensionality reduction.
Also, we prove in Theorem 2 that the sample complexity is completely independent of dimensionality of the data.

**Q7)** "The objective functions in (3) and (4) are based on the expectation of functions defined in line 104 which are not
empirically available."
**A7)** We agree that the objective in (3) cannot be computed. However, the objective in (4) can be computed straight-
forwardly using the data as it involves empirical estimates alone. The interesting result in Theorem 2 shows that at
optimality, (4) converges to (3) at a rate that is $O(1/\sqrt{\min(m,n)})$ and is dimension-free.

**Q8)** "The algorithm discussed in Section 3.4 is based on a couple of simplification conditions. It will be helpful to
discuss the approximation error as well as the computational complexity of the algorithm."
**A8)** Yes, discussion in Section 3.4 is only for special cases ($\epsilon_i = 0$). If we wish to choose hyper-parameters that do
not satisfy these conditions, i.e., if $\epsilon_i \neq 0$, then instead one can always solve the convex problem (5) using existing
off-the-shelf solvers. So there will be no "approximation error" if (5) is solved directly. In supplementary Section 3.4,
we provide details of computational complexity (lines $239 - 243$ and $246 - 247$).

**Q9)** "Simulations on Gaussian OT may not be sufficient to demonstrate the effectiveness of the proposed method"
**A9)** We agree. This is the reason we also demonstrate performance on real-world benchmark problems in domain
adaptation application (Section 5 and Table 1 in the main paper, and Section 5.2 in the supplementary material).

**Q10)** "how to construct $\Sigma_1$ and $\Sigma_2$ in the multivariate Gaussian example?"
**A10)** Additional details of experiments are present in the supplementary material. For instance, lines 279-281 in the
supplementary material discusses how to construct $\Sigma_1$ and $\Sigma_2$.

[Meta-Review · NeurIPS 2020]

The paper addresses the problem of estimating transport map between continuous distributions through kernel mean embeddings. All reviewers agreed that the paper proposes a noveland interesting contribution to the OT literature by bridging kernel mean embeddings and OT. They achieve strong and relevant guarantees (eg dimension-free sample complexity of the transport map estimation). As such, reviewers all think that the paper has to be accepted.